# Direct and indirect neurogenesis from radial glial progenitor cell clones in the mouse neocortex

Dan Shen ⬛, Xin-Yi Wang ⬛, Ruo-Hang Liu, Huan-Huan Deng, Shi-Yuan Tong, Jun-Yang Chen ⬛, Zi-Yun Zhai, Yuan-Xin Li, You-Ning Lin, Fu-Wei Yang, Chen-Xi Wang, Lin-Yun Liu ⬛ ✉, Ying Zhu ⬛ ✉ & Yong-Chun Yu ⬛ ✉

## Abstract

The size and complexity of the neocortex are largely determined during brain development by neurogenesis from radial glial progenitor (RGP) cells. Neurogenesis from such cells can be direct (i.e., RGP cells give rise directly to neurons) or indirect (i.e., RGP cells first produce intermediate progenitor cells, which then divide further to produce neurons). How direct and indirect neurogenesis from RGP cells leads to the appropriate neocortical size and cell-type composition remains incompletely understood. In this study, we developed a combined retrovirus- and FlashTag-based labeling technique that allows clonal tracking of sequential RGP divisions and identification of progeny identities in vivo. Using this method, we show that divisions of mouse RGP cells giving rise to neurogenic (N), neurogenic intermediate progenitor (IP), and neurogenic proliferative intermediate progenitor (IPP) cells tend to generate similar numbers of pyramidal neurons. In the early neuronal progeny of RGP cells, the distribution of neurons produced by the N-, IP-, and IPP-producing divisions follows an "inside-out" pattern in the neocortex. Clonal analysis and mathematical modeling indicate that RGP cells initially give rise to neurons, IP, and IPP cells in a stochastic manner, followed by relatively stable transition patterns between direct and indirect neurogenesis across successive generations. These findings provide a comprehensive and novel understanding of the dynamics of cell division during cortical neurogenesis.

**Keywords** Cell Lineage; Intermediate Progenitor Cell; Neocortex; Neurogenesis; Radial Glial Progenitor Cell
**Subject Categories** Development; Neuroscience; Stem Cells & Regenerative Medicine

## Introduction

The neocortex is responsible for the execution of almost all higher-order brain functions. It consists of millions to billions of highly diverse neurons that are organized into distinct layers to support its functions. Mammalian neocortex development is a complex, and highly coordinated process that is crucial for establishing the brain's intricate structure and functionality. Understanding how the neocortex attains an appropriate size, cell-type composition, and layer formation from a pool of proliferating and differentiating progenitors is a key question in developmental neuroscience.

In the developing neocortex, radial glial progenitors (RGPs) in the ventricular zone (VZ) produce nearly all excitatory neurons, and their proliferation dynamics during temporal lineage progression critically determine the final number and composition of neurons in the mature neocortex (Anthony et al, 2004; Gao et al, 2014; Holguera and Desplan, 2018; Llorca et al, 2019; Malatesta et al, 2000; Miyata et al, 2001; Noctor et al, 2001; Tamamaki et al, 2001). In rodents, RGPs undergo asymmetric divisions to generate excitatory neurons either directly or indirectly via intermediate progenitors (IPs), and at the same time, to self-renew (Haubensak et al, 2004; Miyata et al, 2004; Noctor et al, 2004). Direct neurogenesis produces neurons quickly, but RGPs can only produce one neuron per cell cycle. Compared with direct neurogenesis, neuron production from indirect neurogenesis is slower because it involves a transit step of IP generation, but the final neuron production is greater. Although IPs in the subventricular zone (SVZ) of rodents typically undergo once symmetric division to produce a pair of neurons ("neurogenic"), experimental in vitro and in vivo evidence support the existence of proliferative IPs which give rise to two IPs and finally generating four neurons (Noctor et al, 2004; Postel et al, 2019; Wong et al, 2015). Newborn neurons migrate along the radial glial fibers of mother RGPs to constitute the future neocortex and follow a general "inside-out" distribution, that is, early-born neurons occupy the deep cortical layers and later-born neurons progressively reside in more superficial layers (Angevine and Sidman, 1961; Kornack and Rakic, 1995; Magrinelli et al, 2022; Polleux et al, 1997; Takahashi et al, 1999). In addition, each RGP sequentially produces sibling neurons and forms a radial unit of clonally related excitatory neurons (radial unit hypothesis), sometimes referred to as an ontogenetic column (Rakic, 1988). Despite substantial progress in elucidating the principles of RGP lineage progression (Cadwell et al, 2020; Costa and Muller, 2014; Eckler et al, 2015; Franco et al, 2012; Franco and Muller, 2013; Gao et al, 2014; Gil-Sanz et al, 2015; Kaplan et al,

Jing'an District Central Hospital of Shanghai, Institutes of Brain Science, State Key Laboratory of Brain Function and Disorders, MOE Frontiers Center for Brain Science, Fudan University, 200032 Shanghai, China. ✉E-mail: liuly@fudan.edu.cn; ying_zhu@fudan.edu.cn; ycyu@fudan.edu.cn

2017; Llorca et al, 2019; Qian et al, 1998; Shen et al, 2006), it still remains largely unknown how the precise quantitative (i.e., cell number) and qualitative (i.e., cell type) output of individual RGP clones is modulated and, in particular, how such output is temporally coordinated by direct versus indirect neurogenesis.

Over the past three decades, rapid advances in chemical, genetic, imaging, and sequencing technologies have led to the development of numerous methods for lineage tracing and clonal analysis. Generally, there have been seven major paradigms for studying cell lineages: time-lapse imaging (Noctor et al, 2004; Wang et al, 2011), dye or retrovirus labeling (Cepko et al, 2000; Golden et al, 1995; Price, 1987; Szele and Cepko, 1996), genetic multicolor labeling (Livet et al, 2007; Snippert et al, 2010), DNA barcode-based lineage tracing (Fuentealba et al, 2015; Golden et al, 1995; Harwell et al, 2015; Kirkwood et al, 1992; Lu et al, 2011; Mayer et al, 2015; Sultan et al, 2016; Walsh and Cepko, 1992), CRISPR-Cas9 genome-editing lineage tracing (Crosetto et al, 2015; Frieda et al, 2017; Junker et al, 2017; McKenna et al, 2016; Schmidt et al, 2017), somatic mutation lineage tracing (Ellegren, 2004; Erwin et al, 2014; Evrony et al, 2015; Frumkin et al, 2005; Lodato et al, 2015; McConnell et al, 2013; Muotri et al, 2005; Shapiro et al, 2013), and mouse genetic lineage tracing (Branda and Dymecki, 2004; Huang and Zeng, 2013; Orban et al, 1992). For example, lineage tracing experiments using the powerful Mosaic Analysis with Double Markers (MADM) technology have delineated quantitative information on the patterns of RGP division across the spatiotemporal axis (Amberg et al, 2022; Beattie et al, 2017; Bonaguidi et al, 2011; Gao et al, 2014; Kaplan et al, 2017; Liu et al, 2011; Llorca et al, 2019; Lv et al, 2019; Shen et al, 2021). MADM method relies on the reconstitution of two fluorescent marker genes (i.e., EGFP and tdTomato) in a dividing RGP, which are then segregated into the two daughter cells as well as their progeny, allowing the assessment of the first division pattern of the RGP, rather than the reconstruction of the entire lineage tree (Zong et al, 2005).

In this study, we developed Rv-FlashTag, a retrovirus and dye combination tool which allows precise tracking of sequential cell division events at the single-cell level. Using Rv-FlashTag, we characterized the different cellular identities of the tracked cells and their relationship to the lineage branching, and performed a quantitative clonal analysis of RGP division and lineage progression in the developing mouse neocortex.

## Results

### Lineage tree reconstruction of neocortical pyramidal neurons by Rv-FlashTag

An ideal lineage-tracing experiment should simultaneously capture clonal history, cell identity, and spatial distribution at single-cell resolution, presenting a major challenge in determining the fate of progenitor cells in the adult neocortex. The goal of our experiment is twofold: first, to link the sequence of embryonic progenitor cell division pattern to their corresponding clones in the postnatal neocortex; second, to quantify the cell-type composition of these clones, and thereby determine the multipotency of embryonic progenitors.

To reach the first goal, we developed the Rv-FlashTag, a dual-labeling method to mark clonally related pyramidal neurons (PyN)

and trace their lineage branching. Rv-FlashTag works by the injection of low-titer retroviruses expressing Cre recombinase (RV-Cre) into the lateral ventricle of embryonic day 12 (E12) Ai140 mouse embryos in utero to label dividing RGPs and their progeny with characteristic PyN morphology, followed 15–30 min later by injecting FlashTag (carboxyfluorescein succinimidyl ester, CFSE) into the same ventricle (Fig. 1A). CFSE specifically labels mitotic RGPs adjacent to the ventricle, and its fluorescence drops by half in RGPs and their progenies with each subsequent cell division (Govindan et al, 2018; Jabaudon, 2017; Quah and Parish, 2012; Quah et al, 2007; Telley et al, 2016) (Fig. 1B,C). Consistent with this, CFSE fluorescence in PH3-positive (PH3$^+$) mitotic cells in the VZ and SVZ split nearly by half in the two daughter cells (Fig. EV1A–E). Time-lapse imaging of CFSE-expressing RGPs in the VZ and IPs in the SVZ demonstrated symmetric inheritance of CFSE fluorescence during cell division. Each daughter cell received approximately half CFSE fluorescence of the mother cells, consistent with the theoretically expected values (Fig. EV1F–J). Additionally, CFSE fluorescence localized to the RGP process accounted for ~9.8% of total cellular CFSE fluorescence (Fig. EV2). Given the minimal transfer of process-associated fluorescence to daughter cells during division, this part of fluorescence is unlikely to significantly affect lineage tree reconstruction and analysis.

In general, CFSE fluorescence was higher in cells located in the deep layers (DLs) of the neocortex and lower in those located in the more superficial layers (SLs) (Fig. 1C, right panel). This distribution pattern corresponds to the birthdate-dependent inside-out migration of cortical PyNs and their positioning. It's worth noting that CFSE was primarily expressed in the cell bodies, as previously reported (Quah and Parish, 2012; Quah et al, 2007) (Fig. 1C). Therefore, we measured the integrated CFSE intensity of the soma for each cell within the clone using three-dimensional (3D) confocal images at postnatal day 7 (P7), a time point at which neuronal migration is complete (Magrinelli et al, 2022) (Fig. 1H). To ensure reliable detection of CFSE fluorescence in cells, the fluorescence threshold was set at least twofold above background autofluorescence. To recover the full complement of labeled cells, we performed serial brain sectioning followed by 3D reconstruction of individual clones (Fig. 1F).

To further quantify the cell-type composition of individual clones, we performed immunohistochemical staining for Satb2, Ctip2 and Fog2 at P7 (Fig. 1D). Based on the expression of these markers and the laminar distribution of PyNs, we identified the following cell types: callosal projection neuron (CPN, high expression of Satb2 and negative for Ctip2 and Fog2), subcerebral projection neuron (SCPN, high expression of Ctip2 and negative for Satb2 and Fog2, mainly located in layer 5), corticothalamic projection neuron (CThPN, expression of Fog2, located mainly in layer 6), and heterogeneous projection neuron (HPN, coexpression of Satb2 and Ctip2, located in layer 5) (Arlotta et al, 2005; Galazo et al, 2016; Harb et al, 2016; McKenna et al, 2015; Wuttke et al, 2018) (Fig. 1E,G).

During neurogenesis, RGPs divide asymmetrically to self-renew and produce either one neuron through direct neurogenesis (N division) or one IP through indirect neurogenesis (Miyata et al, 2004; Noctor et al, 2004) (Fig. 1I). IPs can be classified into two principal types: neurogenic IP, which divide symmetrically to produce two neurons, and proliferative IP (IPP), which give birth to two neurogenic IPs, each of which subsequently generates two neurons (Miyata et al, 2004; Noctor et al, 2004; Wong et al, 2015)

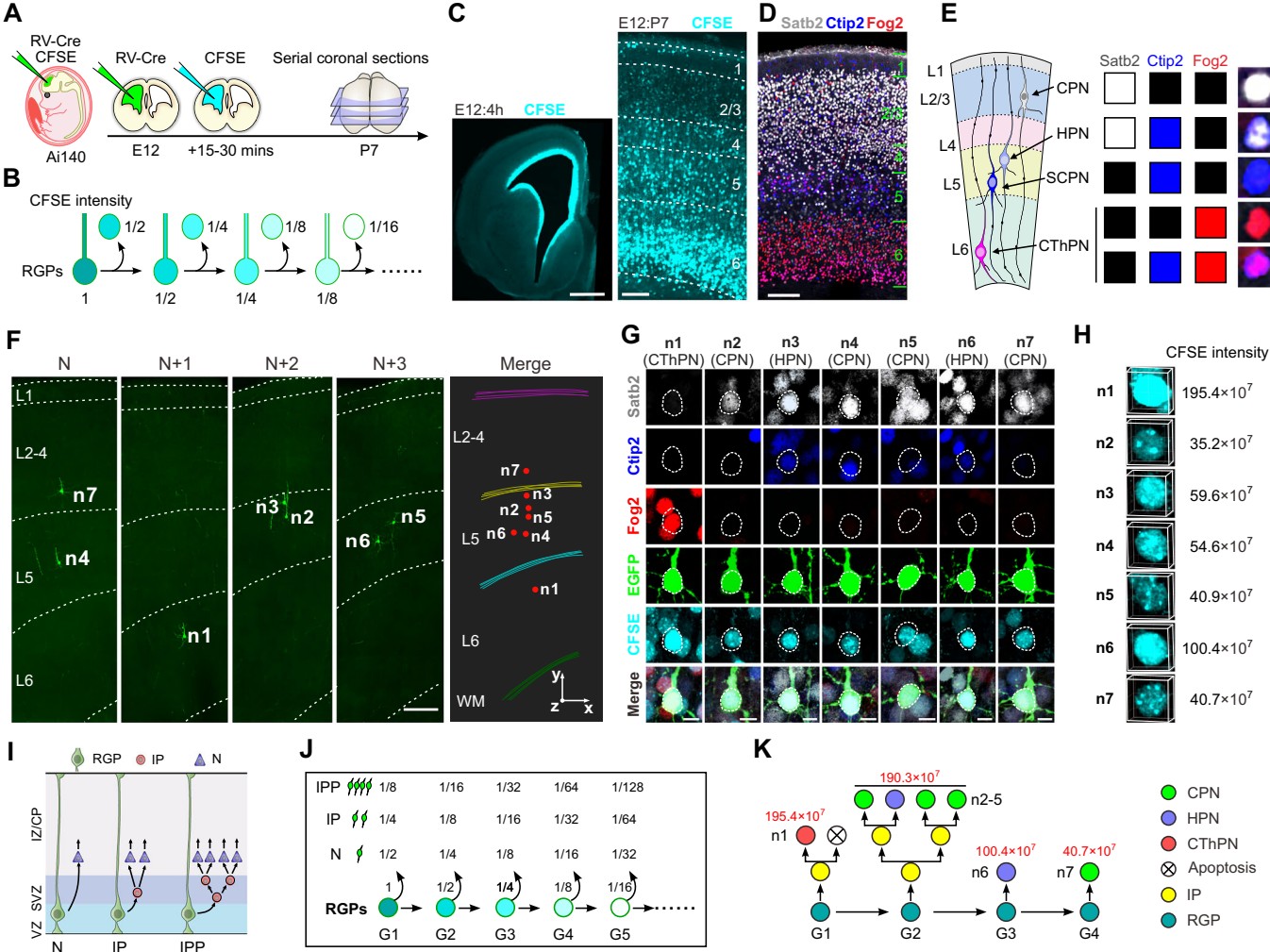

**Figure 1. Rv-FlashTag and lineage reconstruction.**

(A) Experimental paradigm of Rv-FlashTag lineage tracing. (B) In the Rv-FlashTag labeled lineage, CFSE fluorescence intensities halved in RGPs and their daughter cells with each generation. (C) Restrictive distribution of CFSE dye in mouse neocortex 4 h after CFSE injection at E12 (left). Scale bar: 200 μm. Distribution of neurons with different CFSE fluorescence intensities in P7 neocortex following injection at E12 (right). Scale bar: 100 μm. (D) Layer-specific distribution of Satb2, Ctip2, and Fog2 in the P7 cortex. Fog2 is predominantly expressed in L6, Ctip2 mainly in L5, and Satb2 primarily in L4 and L2/3, with weaker expression in L5 and L6. Scale bar: 100 μm. (E) Determination of excitatory neuronal types using Satb2, Ctip2, and Fog2 staining. L layer. (F) Confocal images of a representative lineage containing four generations labeled with Rv-FlashTag at E12 and examined at P7. 3D reconstruction image of the lineage is shown on the right. Red dots indicate the cell bodies of labeled neurons. Colored lines indicate the brain contours and layer boundaries. Layers are shown on the left. n1–n7: neuron number. WM white matter. Scale bar: 150 μm. (G) High-magnification images of all daughter neurons from the lineage in (F). Consecutive brain sections were stained with antibodies against Satb2 (white), Ctip2 (blue), and Fog2 (red). n1 ~ n7 represent neurons in (F). CThPN: corticothalamic projection neuron; CPN: callosal projection neuron; HPN: heterogeneous projection neuron. Scale bar: 5 μm. (H) CFSE fluorescence intensities for neurons n1 ~ n7. (I) Three RGP division patterns including N, IP, and IPP divisions. N neuron, IP intermediate progenitor, IPP proliferative IP. (J) Theoretical changes in CFSE fluorescence intensities across subsequent divisions in the lineage, with the initial RGP intensity set to 1. G1 ~ G5: generations of dividing RGPs. (K) Reconstruction of the lineage in (F) and its progeny subtypes. Red numbers indicate total CFSE intensities for each generation. Source data are available online for this figure.

(Fig. 1I). We defined each division of RGPs and their progenies as a generation (G). It is worth noting that we defined the initially labeled RGP as neurogenesis G1, even though it may contained a small proportion of RGPs that underwent symmetric division or might have missed the first asymmetric neurogenesis (Gao et al, 2014). In each generation, based on division patterns, the theoretical CFSE fluorescence intensity of each cell in the clone can be calculated (Fig. 1J). Additionally, accumulating evidence indicates that IP-derived neurons tend to occupy the same cortical layer and cluster closely in individual clones (Huilgol et al, 2025;

Mihalas and Hevner, 2018; Noctor et al, 2004; Qian et al, 1998). Thus, Rv-FlashTag, in combination with immunohistochemical staining, allows us to reconstruct the lineage tree, identify the location of lineage cells in the neocortex, and obtain a picture of how different types of cells are formed during progenitor divisions in vivo at the single-cell level (Fig. 1K; Table EV1). Of note, in some cases (e.g., shown in Fig. 1K), loss of neurons with corresponding CFSE fluorescence intensities in clones prevented lineage tree reconstruction. While it is unclear what caused the loss of these neurons, we presume that apoptosis was likely to have occurred in

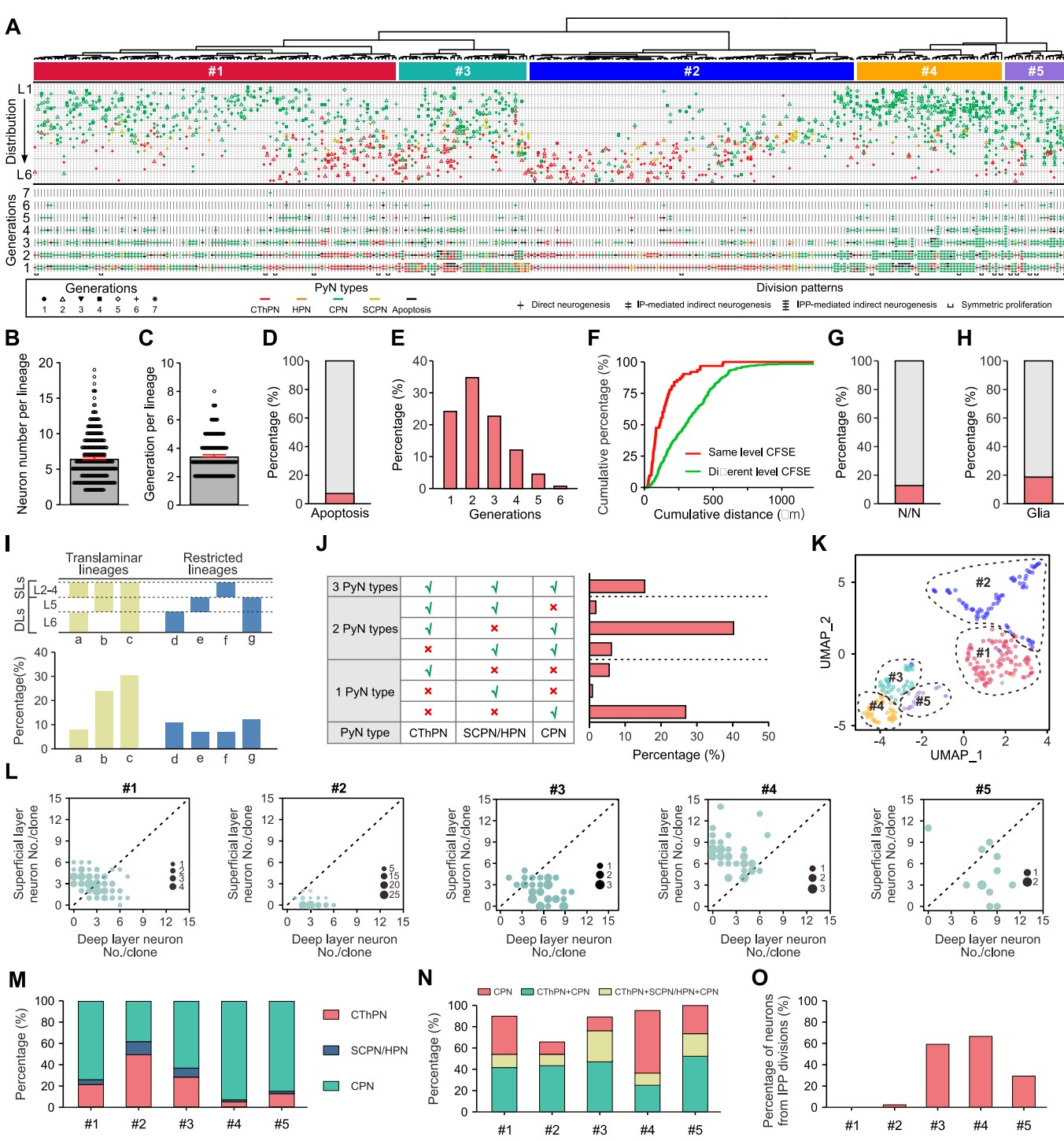

these cells (Gohlke et al, 2004; Kim and Sun, 2011). Moreover, although there is evidence that neurons from the same IP are more closely distributed, this may not always be the case. In particular, the selection of pair neurons originating from the IPs in the IPP division may not accurately reflect the neurogenic process. Given these methodological limitations, our subsequent statistical analyses emphasize generation-level comparisons rather than detailed pairwise lineage assignments.

## Lineage collection and characteristics

Using the Rv-FlashTag method, we obtained 404 lineages from 153 hemibrains (~2–3 lineages/hemibrain), ensuring sparse and well-isolated labelling (Fig. EV3). Of these, 81 lineages contained either only 1, 2, or 4 cells with highly similar CFSE intensities, suggesting that RV-cre labeling occurred in neurons or IPs/IPPs, rather than in RGPs (Cepko et al, 2000; Llorca et al, 2019). These lineages were

**Figure 2.  Characteristics and classification of lineages in the developing neocortex.**

(A) Overview of Rv-FlashTag-labeled lineages arranged by hierarchical clustering, presented in three parts: the top section displays the hierarchical clustering results; the middle section exhibits neuron types, generations, and layer distribution for each lineage; the bottom part illustrates the division patterns across generations within lineages. CThPN corticothalamic projection neuron, HPN heterogeneous projection neuron, SPN subplate projection neuron, CPN callosal projection neuron, SCPN subcerebral projection neuron. (B) Summary of the neuron output per individual lineage ($n = 296$). Data are shown as mean ± SEM. (C) Summary of generations per individual lineage ($n = 208$). Data are shown as mean ± SEM. (D) Summary of neuronal apoptosis percentage in all lineages. Red and gray bars represent the percentages of apoptotic and non-apoptotic neurons in all neurons of lineages, respectively. (E) Quantification of the percentages of lineages containing apoptotic neurons across G1 to G6. (F) Cumulative distance analysis comparing distance between neurons with the similar versus different CFSE fluorescence intensities in lineages. (G) Percentage of lineages in which RGP consumes itself to generate two neurons (red bar). (H) Percentage of lineages generating glial cells (red bar). (I) Top: layer distribution analysis classifying clones into three subtypes of translaminar clones (a–c) and four subtypes of spatially restricted clones (d-g). Bottom: corresponding percentages of a~g clones in all clones. (J) Percentages of clones generating 1, 2, and 3 PyN types. (K) UMAP visualization of the five clusters identified by hierarchical clustering in (A). (L) Relative abundance of excitatory neurons in superficial versus deep layers for clones in clusters #1 ~ #5. Circle size represents the number of lineages and its location indicates the number of neurons in deep versus superficial layers. Dashed lines indicate an equal number of neurons in both deep and superficial layers within a clone. (M) Summary of PyN subtype percentages in lineage clusters #1 ~ #5. (N) Summary of the three main PyN subtype combinations in lineage clusters #1 ~ #5, related to (I). (O) Percentage of neurons derived from IPP divisions in lineage clusters #1 ~ #5. Source data are available online for this figure.

excluded from further analyses. To reconstruct lineage trees, we used automated computational reconstruction followed by manual validation (Fig. EV4, "Methods"), ensuring that each selected lineage generated a single lineage tree. The complete dataset of lineages, including generations, division patterns, cell-type composition, and spatial distribution within the neocortex, is presented in Fig. 2A. Of note, CFSE fluorescence did not distribute exactly equally between daughter cells during each division in lineage tree construction, but instead exhibited deviations that followed a normal distribution pattern (Figs. 1H,K and EV5A). To evaluate how these observed fluorescence deviations might impact lineage tree reconstruction accuracy, we generated 100 in silico lineage trees and assigned CFSE fluorescence values to daughter cells by randomly sampling deviations from the experimentally observed normal distribution (Fig. EV5B–D). Quantitative analysis revealed that all perturbed lineage trees were reconstructed with perfect accuracy, identical to the structure of their unperturbed counterparts, demonstrating that the observed biological deviations in CFSE fluorescence distribution do not compromise the accuracy of automated lineage tree reconstruction (Fig. EV5E,F).

In the developing mouse neocortex, RGPs transit from symmetric proliferation to asymmetric neurogenic divisions at E11–E12 (Haubensak et al, 2004). Indeed, we observed that symmetric lineages represented ~7% of the lineages obtained at E12 (20/302) (Fig. 2A). Consistent with previous reports using alternative methods (Gao et al, 2014), this result indicates that, at this stage, most RGPs begin undergoing asymmetric divisions. Quantifying the size of asymmetric clones revealed that individual RGPs generated diverse neurons with a wide range of clone sizes (Fig. 2B), suggesting heterogeneous neuronal outputs. Similarly, we observed a substantial variability in the number of divisions undergone by RGPs. About 79% of RGPs divided only 2-4 times, suggesting a relatively limited number of divisions to generate a massive number of neurons (Fig. 2C). Meanwhile, ~7% of PyNs in clones showed programmed cell death during development (Gao et al, 2014), with the peak occurring at G1-3 (Fig. 2D,E). To quantitatively assess the spatial distribution of IP-derived neurons in clones, we used cumulative distance analysis to compare the spatial distances among neurons with similar versus different CFSE intensities. Although neurons with similar CFSE intensities in clones partially contained some neurons with different generations, the distances among neurons with similar CFSE intensities were significantly shorter than those with different CFSE intensities

(Fig. 2F), supporting the notion that IP-derived neurons tend to cluster closely in individual clones (Mihalas and Hevner, 2018; Qian et al, 1998). Furthermore, ~14% of lineages involved terminal RGP divisions, in which two daughter neurons were generated simultaneously (N/N division) (Miyata et al, 2004; Takahashi et al, 1996) (Fig. 2G). Besides neurons, some RGPs also produce glial cells, including astrocytes and oligodendrocytes (Anthony et al, 2004; Campbell and Götz, 2002; Shen et al, 2021). Consistent with previous report (Gao et al, 2014), we found that ~20% of spatially isolated clonal clusters contained both neurons and glia cells (Figs. 2H and EV6).

We next examined the laminar distribution of clonal neuronal outputs. Consistent with classical models of cortical neurogenesis, we found that most RGPs (~63%) infected at E12 produced translaminar lineages containing neurons in both deep and superficial layers of the neocortex (Fig. 2I,J). However, we also observed a substantial fraction of lineages in which neurons were confined to either DLs or SLs (Fig. 2I,J). In addition, ~35% of the lineages were composed exclusively of one type of PyNs, while multiple combinations of PyN identities comprised the remaining lineages (Fig. 2J). Approximately 16% of the lineages contained the entire complement of PyN subtypes identified and ~40% included both CPNs and CThPNs. These results suggest that RGP neuronal outputs are both constrained and heterogeneous.

## Lineage classification

To quantitatively assess the properties of individual clones, we extracted 20 features from each lineage. These quantitative features represented multiple aspects of the lineages, including division patterns, laminar distribution, and excitatory neuron types, that could be robustly extracted and compared across all lineages (Table EV2). These features provided the foundation for an unsupervised classification of lineages. Using hierarchical Ward's cluster analysis, we identified 5 broad clusters (Fig. 2A). Using uniform manifold approximation and projection (UMAP) to project the high-dimensional data into two dimensions, we observed that lineages assigned to different clusters were generally localized in distinct regions in this space (Fig. 2K), indicating that each cluster possesses unique properties that set it apart from the others. For example, by quantifying the relative proportions of deep and superficial layer neurons, we found that lineages in clusters 2 and 3 predominantly produced DL neurons, whereas those in

cluster 4 showed a significant bias towards producing SL neurons (Fig. 2L). In contrast, cluster 1 displayed a relatively balanced distribution across deep and superficial layers (Fig. 2L). Accordingly, these five clusters showed significant differences in their constituent neuronal subtypes (Fig. 2M) and three main combinations of PyN subtypes in lineages (only CPN, CThPN+CPN, and CThPN+SCPN/HPN + CPN) (Fig. 2J,N). Furthermore, the generation of IPPs and subsequent indirect neurogenesis contributed 59.8%, 67.3% and 29.6% of neurons in clusters 3, 4 and 5, respectively, but were rarely observed in clusters 1 and 2 (Fig. 2O). Additionally, these five clusters also exhibited distinct spatial distributions across the neocortex (Fig. EV7A). For example, cluster 2 was significantly enriched in the auditory cortex (AC) compared to the motor cortex (MC), the somatosensory cortex (SC), and the visual cortex (VC), whereas cluster 4 showed more prominent in the MC than in the other three regions (Fig. EV7A). These regional distribution patterns suggest that clonal properties may be influenced by local cortical architecture and developmental microenvironments (Gao et al, 2014). Notably, neither clone size nor generation number showed significant differences across the four cortical areas (Fig. EV7B,C). Taken together, these results suggest that RGPs exhibit distinct behaviors and properties at the clonal level.

## Direct versus indirect neurogenesis

The balance between direct and indirect neurogenesis is crucial for generating the appropriate numbers and types of neurons (Mihalas et al, 2016). Thus, we further took advantage of the Rv-FlashTag method in distinguishing the progenitor origin of daughter cells in lineages and quantitatively compared the neuronal outputs and neocortical distributions between direct and indirect neurogenesis. As RGP division proceeded, we found that the number of PyNs produced by the RGPs decreased exponentially (Fig. 3A). 86% of neurons across all lineages were produced by the first three generations of RGPs (Fig. 3A). Furthermore, the proportions of RGPs producing IPPs, IPs, and neurons (N division) were 12.5%, 28.4%, and 59.1%, respectively (Fig. 3B). Notably, as development progressed, the proportions of IPP divisions decreased dramatically, whereas the proportions of N divisions progressively increased (Fig. 3C). In contrast, the proportions of IP divisions remained relatively stable throughout the development, with a decline only in the final division (Fig. 3C). Strikingly, we found that the ratio of PyNs produced by the IPP, IP and N divisions of RGPs was approximately 1:1:1 (Fig. 3D). More specifically, the N, IP, and IPP divisions of RGPs tended to produce similar proportions of the four identified PyN subclasses (CThPN, SCPN/HPN, DL-CPN and SL-CPN) (Fig. 3E,F). Interestingly, we observed that, as RGP division proceeded, the number of CThPN, DL-CPN, and SCPN/HPN produced by the RGPs decreased exponentially, but the production of SL-CPN showed Gaussian distribution (Fig. EV8). Together, these results reveal that the neuronal outputs of both direct and indirect neurogenesis in development are constrained and balanced at a fine level.

Next, we focused on the laminar fate of sequential generations of PyNs. In addition to an overall inside-out lamination pattern, early-generation neurons were distributed broadly in both DLs and SLs, whereas later-generation neurons exhibited more compact laminar distributions, and mainly occupying SLs (Fig. 3G). These results

suggest that early neurogenic divisions of RGPs produce neurons with heterogeneous laminar fates, whereas later on, fate control becomes tighter and PyN identity is more homogeneous. Indirect neurogenesis not only increases the neuronal output, but also affects the layer positioning of neurons (Lv et al, 2019; Mihalas et al, 2016). Since additional round(s) of cell divisions via IPs/IPPs postpone the actual birthdate of neurons after the initial RGP division, they may consequently influence their laminar positioning. To examine this hypothesis, we systematically analyzed the neocortical distributions of neurons produced by direct and indirect neurogenesis across successive RGP generations. We observed that in early generations of RGPs (G1-3), the mean relative radial positions of PyNs produced by the N, IP and IPP divisions followed an "inside-out" pattern (Fig. 3H). Specifically, neurons produced by N, IP and IPP divisions were respectively located from lower to upper radial neocortical positions. However, this pattern was not observed in late RGP generations (G4-6) (Fig. 3H). These results strongly suggest that during early neurogenesis, indirect neurogenesis, compared to direct neurogenesis, generates neurons that tend to occupy upper radial neocortical positions, leading to heterogeneous laminar fates of early RGP neuronal outputs.

Despite the existing evidence and Rv-FlashTag lineage tracing supporting the existence of IPPs (Miyata et al, 2004; Noctor et al, 2004; Wong et al, 2015), their specific contribution to neurogenesis remains unknown. To directly observe IPP divisions, we prepared organotypic slice cultures from RV-cre-injected Ai140 embryos and performed long-term time-lapse imaging to distinguish direct and indirect neurogenesis in individual clones between E13-E16 (Fig. 4A). We monitored the fates of 74 RGP divisions from 38 asymmetric-division clones. Indirect neurogenesis was identified when a daughter cell generated by RGP division in the VZ subsequently moved to the adjacent SVZ and divided. We found that 56.8% of RGP divisions directly generated PyNs (42/74) (Fig. 4B,C,H; Movie EV1). 23.0% of RGP divisions resulted in an asymmetric fate for IPs that produced two daughter cells and migrated away from the SVZ (17/74) (Fig. 4D,E,H; Movie EV2). Importantly, we found that 13.5% of RGP divisions produced IPPs (10/74). These IPPs generated pairs of new IPs by symmetric division and subsequently divided into four cells in the SVZ (Fig. 4F–H; Movie EV3). In 6.8% of RGP divisions, we were unable to classify the division pattern (Fig. 4H). Notably, the proportion of RGPs producing IPPs, IPs, and neurons was approximately the same as those observed in Rv-FlashTag labeling experiments in vivo (Figs. 3B and 4H). No oRG-like cells were observed in 74 RGP divisions (Wang et al, 2011).

We further examined how cell-cycle duration influences PyN birth time. The averaged cell-cycle lengths were 18.08 ± 1.74 h for RGPs generating postmitotic neurons directly ($t_1$), 15.00 ± 4.60 h for RGPs producing IPs ($t_2$), and 16.95 ± 2.75 h for IPs generating postmitotic neurons ($t_3$), with no significant differences among these three groups (Fig. 4I,J). In contrast, the total time required for IPPs to produce postmitotic neurons after undergoing two rounds of divisions ($t_4$) was significantly extended to 32.75 ± 5.24 h, nearly double that of $t_3$ (Fig. 4I,J). Notably, the residence time of postmitotic neurons in the SVZ (i.e., the time between a neuron's production and its departure from the SVZ) showed no significant differences among postmitotic neurons derived from RGPs ($T_1$), IPs ($T_2$), or IPPs ($T_3$) (Fig. 4K,L). These results suggest that

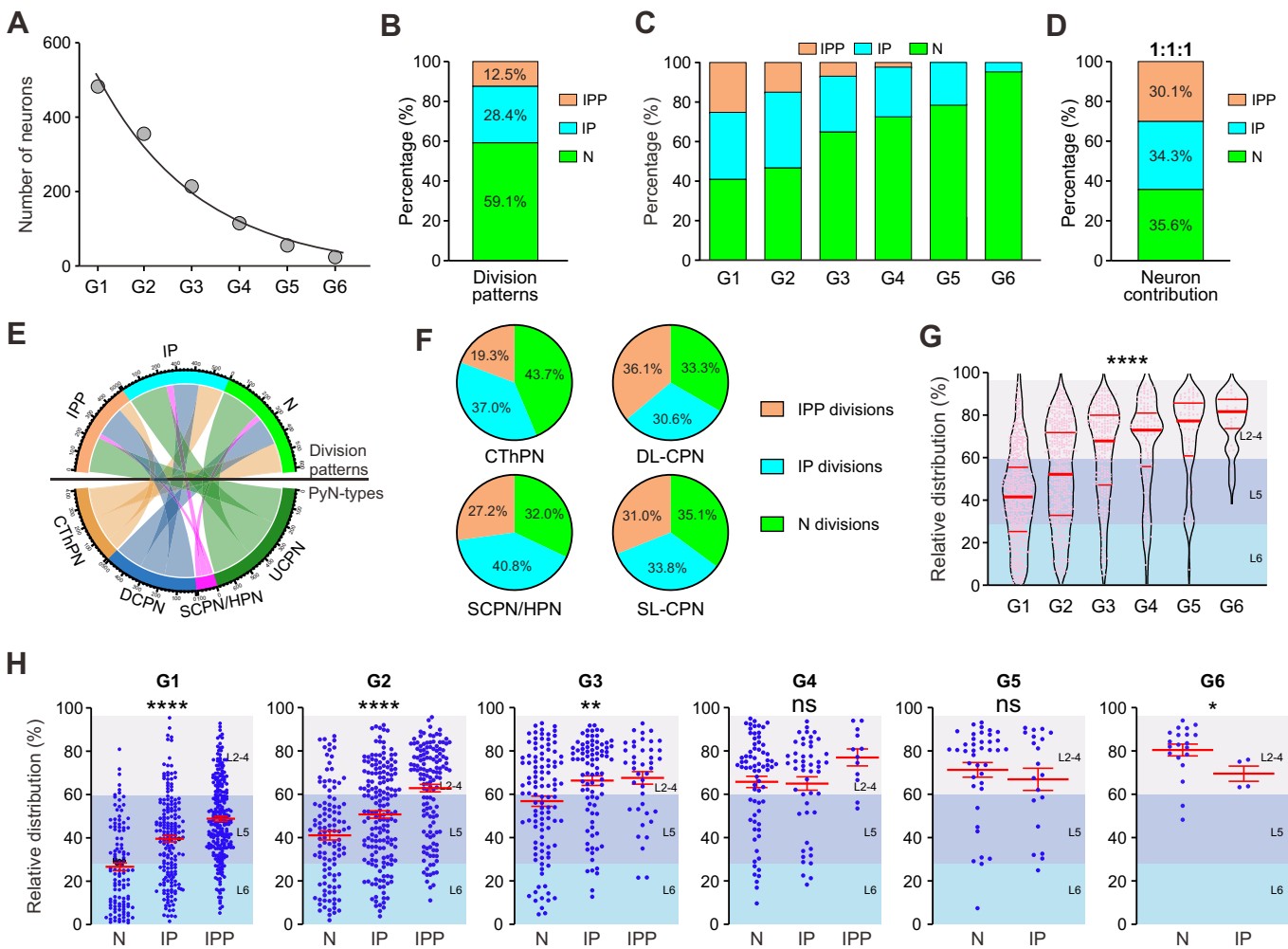

**Figure 3. Direct and indirect neurogenesis in the developing neocortex.**

(A) PyN outputs in G1-G6. The mean of PyN outputs decreases exponentially across generations. Decay index $\lambda = -0.4898$. (B) Percentages of IPP, IP, and N divisions in all lineages. (C) Percentages of PyNs generated by IPP, IP, and N divisions across G1 to G6. (D) Percentages of PyNs generated by IPP, IP and N divisions in all lineages. (E) Relationships between division (IPP, IP and N patterns divisions) and PyN subtypes (CThPN, DL-CPN, SCPN/HPN, SL-CPN) are shown in the circular Sankey diagram. (F) Contributions of IPP, IP, and N divisions to four PyN subtypes. (G) Distribution of PyNs in the neocortex across G1 to G6 (from left to right, $n = 553, 440, 261, 136, 70, 24$). Red horizontal lines indicate median and quartile values. Cortical layers are shown on the right. Data are shown as mean ± SEM. ****$P < 0.0001$ (Kruskal–Wallis one-way ANOVA). (H) Cortical distribution of clonal neurons generated by IPP, IP, and N divisions across G1 to G6 (from left to right, G1: $n = 117, 183, 253$; G2: $n = 113, 174, 139$; G3: $n = 107, 86, 41$; G4: $n = 72, 48, 12$; G5: $n = 40, 20$; G6: $n = 20, 4$). Cortical layers are shown on the right. Data are shown as mean ± SEM. G1: ****$P < 0.0001$ (Kruskal–Wallis one-way ANOVA). G2: ****$P < 0.0001$ (Kruskal–Wallis one-way ANOVA). G3: **$P = 0.0058$ (Kruskal–Wallis one-way ANOVA). G4: $P = 0.2213$ (Kruskal–Wallis one-way ANOVA). G5: $P = 0.6891$ (unpaired Student's $t$ test followed by Mann–Whitney test). G6: $P = 0.0292$ (unpaired Student's $t$ test followed by Mann–Whitney test). Source data are available online for this figure.

additional rounds of division via IPs/IPPs delay the birthdate of PyNs upon RGP division, and potentially affecting their final radial positioning in the neocortex.

## Stable transition probabilities of RGP division patterns across generations

During cortical development, RGPs dynamically switch between generating neurons, IPs and IPPs. Balancing the temporal transition of RGPs between direct and indirect neurogenesis is essential for precisely regulating the number of neurons in the neocortex (Mihalas et al, 2016). We next systematically studied the transitions of RGP division patterns between generations. In the

first generation of RGP neurogenesis, we found that the proportions of RGPs producing neurons, IPs and IPPs were 40.7% (105/258), 33.7% (87/258) and 25.6% (66/258), respectively, indicating that RGPs seem to randomly give rise to neurons, IPs and IPPs in G1 (Fig. 5A,B). As neurogenesis progressed, 25.9-50.0% of RGPs changed their division patterns between consecutive intergeneration, and the proportions of RGPs exiting the neurogenic cycle significantly increased (Fig. 5C,D). Notably, the overall transition probabilities of RGP division patterns remained largely consistent across generations (Fig. 5A). We calculated the transition probabilities for the first four generations of RGPs, which were the main generations to generate neurons. Chi-square tests assessing the dependency of division patterns between two

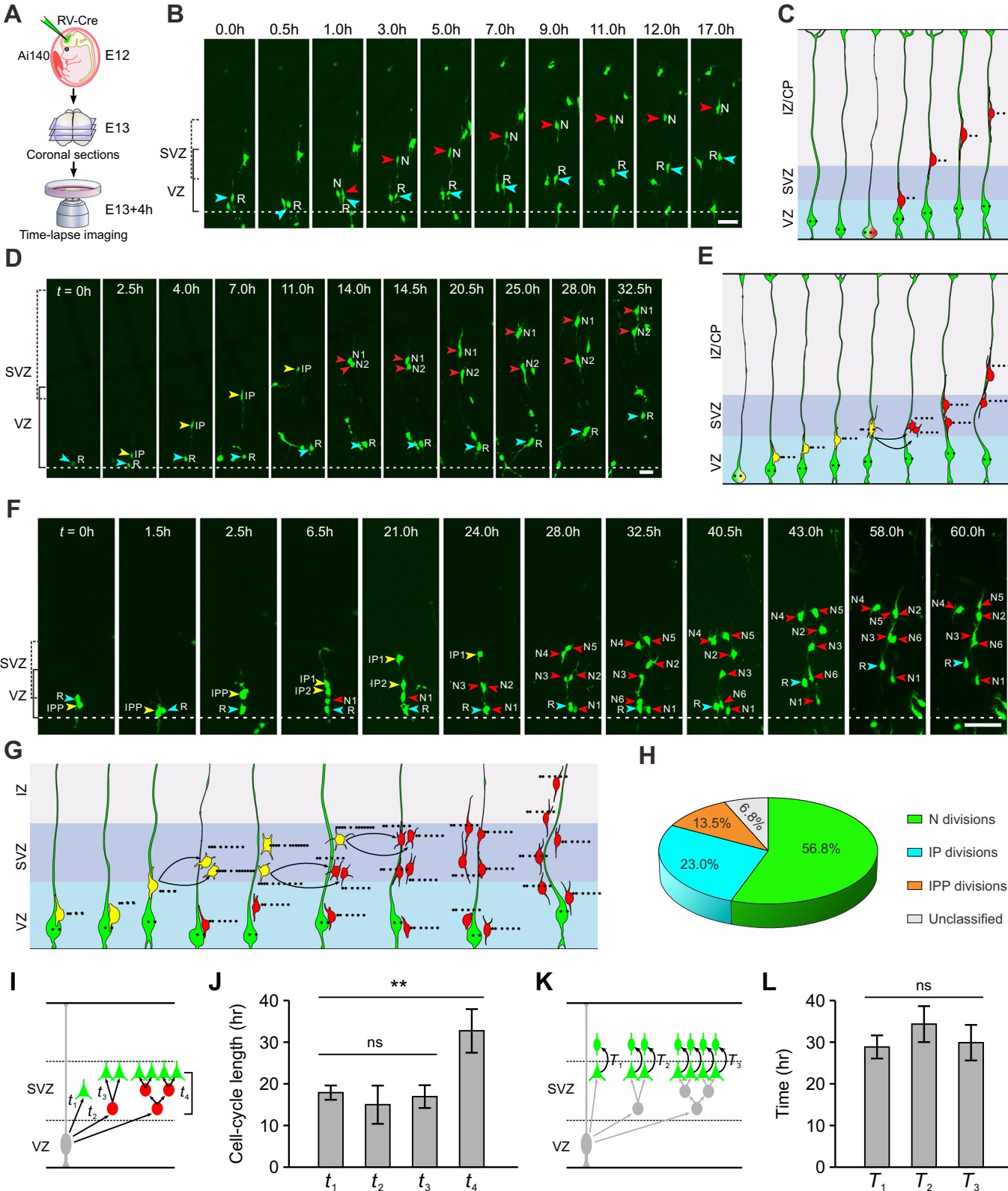

**Figure 4.   Time-lapse imaging of RGP clones.**

(A) Experimental paradigm for lineage tracing with time-lapse imaging. (B, C) Time-lapse imaging for ~72 h and schematic of N division of an RGP. A neurogenic RGP (R, cyan arrowheads) divided asymmetrically in the VZ to self-renew and directly produce a neuron (N, red arrowheads) which migrated radially to the IZ/CP after birth. Scale bar: 100 µm. (D, E) Time-lapse imaging for 72 h and schematic of IP division of an RGP. An RGP (R, cyan arrowheads) generated a neurogenic IP (yellow arrowheads) that divided once in the SVZ and produced two closely distributed daughter neurons (N, red arrowheads). Scale bar: 20 µm. (F, G) Time-lapse imaging for 72 h and schematic of IPP division of an RGP. An RGP(R, cyan arrowheads) divided in the VZ to self-renew and produce a proliferative IP that then divided to produce two IPs (yellow arrowheads). The two IPs divided symmetrically in the SVZ and produced four progenies (N, red arrowheads). Scale bar: 50  µm. (H) Quantification of percentages of RGP division patterns in the time-lapse experiments. (I, J) Cell cycle lengths of RGPs generating a neuron ($t_1$, $n = 25$) or an IP ($t_2$, $n = 4$), IPs generating two neurons ($t_3$, $n = 10$), and IPPs generating four neurons ($t_4$, $n = 10$). Data are shown as mean ± SEM. $t_1$-$t_3$: $P = 0.7890$ (one-way ANOVA). $t_1$-$t_4$: **$P = 0.0028$ (one-way ANOVA). (K, L) The residence time from birth to the initiation of migration for neurons generated by N division ($T_1$, $n = 10$), IP division ($T_2$, $n = 3$), and IPP division ($T_3$, $n = 5$). Data are shown as mean ± SEM. $T_1$-$T_3$: $P = 0.6560$ (one-way ANOVA). Source data are available online for this figure.

consecutive generations revealed $P$ value was less than $10^{-6}$, indicating that RGP division patterns in one generation are influenced by the division patterns of the previous generation. Transition probabilities were then calculated in each intergeneration. For RGPs undergoing N division and IP division in the previous generation, the probabilities in the subsequent generation were stable and independent of generations ($p = 0.5087$ for N division and $p = 0.5816$ for IP division, Fig. 5E). On the other hand, for IPP divisions, transition probabilities varied significantly between generations ($P = 4.12 \times 10^{-8}$, Fig. 5E), likely reflecting the relatively low proportion of IPP divisions in the lineages, which introduced considerable randomness. Together, these results suggest that the division patterns of RGPs between generations are precisely regulated.

## Cell lineage tree model of neurogenesis

Having identified several characteristic features of lineage trees and the behavior of RGPs, we next made simplifying assumptions to create a tractable mathematical model while retaining the most important aspects of neurogenesis. To this end, we simulated lineage trees using a model based on the following four basic rules derived from our experimental data (Fig. 5F, "Methods"):

1. Clonal size follows a Poisson-like distribution (Llorca et al, 2019; Shen et al, 2024). Analysis of our neurogenic lineages revealed a bimodal Poisson distribution of clonal size with peaks centered at 4.09 and 7.62 cells (Llorca et al, 2019).
2. In G1, RGPs randomly generate neurons, IPs and IPPs.
3. The transition probabilities of the division patterns of RGPs remain constant between generations. The transition probabilities were calculated from the average transition probabilities of the first four generations of RGPs.
4. When the clonal size or the remaining number of PyNs ($PyN_{remain}$) after the sequential divisions is less than four, RGPs neither generate IPPs ($2 \leq PyN < 4$) nor IPs and IPPs ($PyN = 1$).

The model randomly simulated 258 lineage trees per batch and was repeated 100 times. The in silico lineages generated using this model were then compared with the experimental lineages. We found that this model accurately reproduced the main experimental features in our data: (1) The distribution of the number of generations (depth of lineage tree) in in silico lineages was consistent with experimental lineages (Fig. 5G); (2) With increasing generations, the total number of neurons produced by each generation of RGPs exhibited an exponential decline (decay index $\lambda = -0.3759$ for the in silico lineages, Fig. 5H; $\lambda = -0.4898$ for the experimental lineages, Fig. 3A); (3) The relative proportions of N, IP and IPP division patterns in each of the first four generations of RGPs were consistent (IPP division was not observed in the experimental lineages after G5, Fig. 5I); (4) Remarkably, the in silico lineages faithfully replicated the ratio of the IPP, IP and N divisions of RGPs and the ratio of PyNs produced by the IPP, IP and N divisions (~1:1:1) to those measured experimentally (Fig. 5J). Together, our modeling results suggest that the stable transition probabilities between direct and indirect neurogenesis across generations, combined with a bimodal Poisson distribution of clonal sizes, are sufficient to recapitulate the fundamental patterns of RGP-derived neurogenesis observed experimentally.

## Discussion

In this study, we developed Rv-FlashTag to reconstruct the RGP lineage trees of neurogenesis in vivo. Rv-FlashTag provides a quantitative assessment of direct and indirect neurogenesis of RGPs in clones and reveals previously unknown RGP behaviors. Using Rv-FlashTag, we identified five distinct clonal clusters based on multiple features of lineages, suggesting that RGPs exhibit heterogeneous behaviors during development. Our results demonstrate that N, IP and IPP divisions of RGPs tend to generate similar numbers of pyramidal neurons, and the locations of neurons produced by the N, IP, and IPP divisions in the same generation follow an inside-out pattern in the neocortex. In addition, clonal analysis and mathematical modeling provide the first evidence that RGPs randomly give rise to neurons, IPs and IPPs in the first neurogenic generation and later with relatively stable transition probabilities between direct and indirect neurogenesis. Since we analyzed over 300 clones, the features that we discovered concerning RGP behavior and neurogenesis likely reflect the fundamental principles underlying neocortical excitatory neuron production and organization.

As with any methodology, the Rv-FlashTag system has its strengths and weaknesses. One of its strengths is that it enables the quantification of fluorescence intensity, thereby allowing a precise assessment of cell division histories of RGPs at a single cell level. Second, Rv-FlashTag can capture multiple features of lineages in combination with cell type marker staining, thereby allowing a precise assessment of clonal subclusters. As for its limitations, Rv-flashTag is unable to determine the specific time at which neurons are generated in the lineage. In other words, neurons from the same generation or cell division stage do not necessarily indicate that

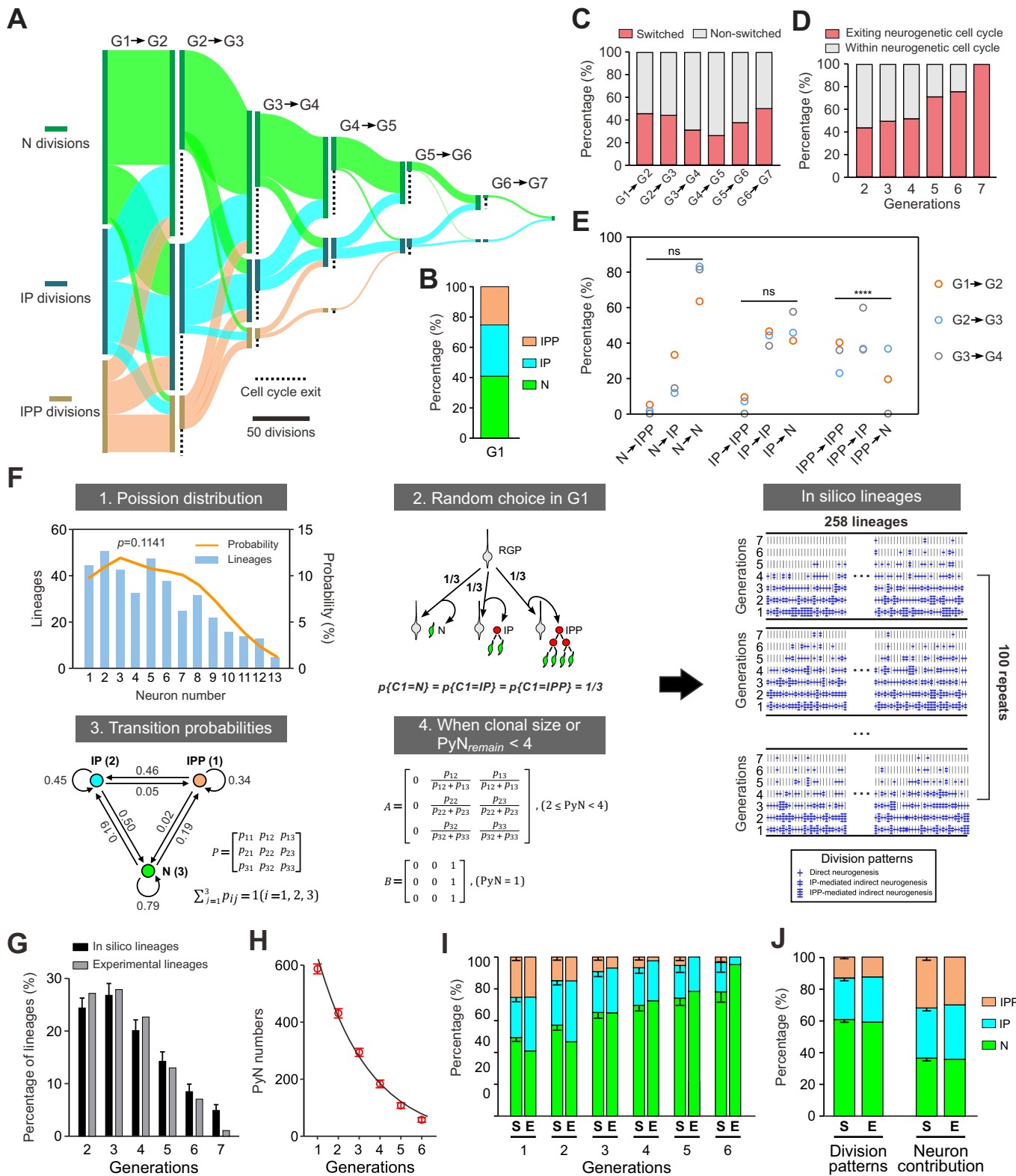

**Figure 5. A model of cortical neurogenesis.**

(A) Sankey diagram shows the transitions of division patterns across generations in lineages. The dashed vertical lines indicate RGPs exiting the cell cycle. Scale bar: 50 RGP divisions. (B) Percentages of IPP, IP and N divisions in the first generation. (C) Percentages of lineages that switched division patterns between two adjacent generations. (D) Percentages of RGPs exiting the cell cycle in G2-G7. (E) Independence on the transition probabilities of RGP division patterns. $P = 0.51$, $P = 0.58$, and ****$P < 0.0001$ (Chi-square test). (F) Workflow of simulated lineage generation. Rule 1 states that clonal size fits a bimodal Poisson distribution. The $P$ value for the chi-square test is 0.2421. Rule 2 states that RGPs randomly choose division patterns in the first generation. Rule 3 states that transition probabilities of RGP division patterns between generations are constant. Rule 4 sets the probabilities of RGP division patterns when the clonal size or the number of PyN remaining ($PyN_{remain}$) after remain sequential divisions is less than four. The model randomly simulated 258 lineage trees and was repeated 100 times. (G) Distribution of RGP generation numbers in experimental ($n = 258$) and in silico lineages ($n = 258$). Data was presented as mean ± SD. Kullback–Leibler (KL) divergence score was used to compare experimental and in silico lineages. KL: 0.0465. (H) The mean of PyN outputs of in silico lineages decrease exponentially across generations ($n = 258$). Decay index $\lambda$ = -0.3759. Data was presented as mean ± SD. (I) The proportions of N, IP and IPP divisions of RGPs in experimental ($n = 258$) and in silico lineages ($n = 258$). S in silico lineages, E experimental lineages. Data was presented as mean ± SD. KL divergence score was used to compare experimental and in silico lineages. KL: 0.0972. (J) The proportions of the IPP, IP and N divisions of RGPs (left) and the proportions of PyNs produced by the IPP, IP and N divisions (right) in experimental ($n = 258$) and in silico lineages ($n = 258$). S in silico lineages, E experimental lineages. Data was presented as mean ± SD. KL divergence score was used to compare experimental and in silico lineages. KL: 0.0019. Source data are available online for this figure.

they are generated in the same embryonic period. Second, the accuracy of clonal analysis with Rv-FlashTag is highly dependent on the initial CFSE uptake of RGPs. Indeed, we observed that some later-generated PyNs in certain lineages exhibited very low or undetectable CFSE fluorescence. These lineages were excluded from subsequent analyses. Moreover, to minimize the effects of variations in conditions, every aspect of the pipeline, from RV-cre and CFSE injection, slice preparation, to staining, imaging, mapping of cells to a reference atlas, and reconstruction, employed highly standardized and quality-controlled methodologies. Third, manually reconstructing lineage trees based on CFSE dilution assay may introduce artificial biases. To minimize bias and increase accuracy, we combined computer analysis with manual checks to reconstruct the lineage trees. The clones that were reconstructed to produce more than one lineage tree were excluded from subsequent analyses. Finally, although retroviral labeling indiscriminately targets dividing progenitor cells, retrovirus silencing can subdue the detection of transfected cells. RV-cre/Ai140 mouse strategy effectively prevents retroviral silencing in dividing RGPs. Although Rv-FlashTag entails several limitations, this framework for tracing cell fate across progenitor cell divisions offers a powerful tool for reconstructing cell lineage trees of neurogenesis and should be broadly useful in stem cell biology.

Our findings indicate that PyN lineages exhibit great heterogeneity in the number, location, and identities they comprise. Using unsupervised clustering methods, we identified five distinct clonal clusters based on multiple lineage features. It should be noted that new features outside the scope of this dataset and/or increased data size could revise the classifications presented here. The underlying molecular and cellular mechanisms resulting in heterogeneous RGP lineages remain unknown. A few major scenarios could be considered. First, programmed cell death of specific PyNs might contribute to the heterogeneous organization of cortical lineages. However, analysis of RGP lineages at P2, before the period of PyN programmed death (Wong et al, 2018), showed that the diversity of lineage patterns was comparable between P2 and P21, although some subtle differences were observed (Llorca et al, 2019). These results suggest that although cell apoptosis may slightly impact the final diversity of lineages, their heterogeneity is likely to arise mainly during the process of cortical neurogenesis. Second, lineage heterogeneity could result from the existence of different RGP "subtypes" with different inherent cellular output potentials. For example, it has been suggested that CUX2$^+$ RGPs are destined to

produce primarily superficial layer neurons (Franco et al, 2012; Gil-Sanz et al, 2015). This idea was challenged by a series of clonal lineage tracing studies showing that individual RGPs labeled at the onset of neurogenesis (~E12) generate both deep and superficial layer neurons (Eckler et al, 2015; Gao et al, 2014; Guo et al, 2013). Interestingly, using an intersection-subtraction strategy with Lhx2-CreER and Fezf2-Flp mouse lines, Matho et al identified distinct RGP subpopulations in the dorsal pallium based on Lhx2 and Fezf2 expression (Matho et al, 2021). Their lineage tracing revealed that RGPs$^{Lhx2+/Fezf2−}$ specifically generate callosal PyNs, while RGPs$^{Lhx2+/Fezf2+}$ produce subcortical PyNs, demonstrating fate-restricted RGP lineages generating distinct neuronal projection classes. Moreover, numerous recent studies using single-cell RNA sequencing (scRNA-seq) approaches have revealed the diversity of RGPs (Di Bella et al, 2021; Li et al, 2023; Li et al, 2020). However, it is important to note that these studies were based on computational methods and confirmation of RGP subtypes in vivo at the cellular level requires future studies. Third, variations in dynamic RGP state transitions during temporal progression may result in lineage heterogeneity. Fourth, variable IP production and/or IP proliferation dynamics could specifically amplify RGP output at defined temporal windows, and thereby within specific layers.

Previous studies have established that early-generated neurons exhibit considerable heterogeneity and are broadly distributed across both deep and superficial layers of the neocortex, whereas late-generated neurons are predominantly restricted to the superficial layers (Frantz and McConnell, 1996; Telley et al, 2016). Building upon this foundational model, our findings demonstrate that indirect neurogenesis is strongly associated with this observed heterogeneity in neuronal distribution patterns. We found that PyNs produced by N, IP, and IPP divisions were sequentially localized from lower to upper radial neocortical positions in each generation of early RGPs, following an inside-out pattern. These results suggest that early indirect neurogenesis, particularly via IPPs, produces heterogeneous laminar fates. Supporting this view, previous genetic fate mapping of Tbr2$^+$ IPs revealed a significant cohort of early IP-derived neurons (~17%) that occupied superficial cortical layers (Mihalas et al, 2016). Using Tbr2-2A-CreER;Ai14 mice, Huilgol et al found that while the overall population of IP-derived PyNs exhibits a deep-to-superficial migration trend, the laminar deployment of successive IP-derived PyN subpopulations markedly deviates from a simple inside-out rule. Specifically, PyNs destined for 2 or 3 non-consecutive layers are generated simultaneously, while different PyN types within the same layer are generated at non-contiguous developmental time points (Huilgol et al, 2024). The

contribution of indirect neurogenesis also varies spatially across brain regions. Along the cortical medial-lateral axis, indirect neurogenesis makes progressively lower contributions, with sharp decreases in the amygdala and piriform cortex (Huilgol et al, 2023). Notably, direct and indirect neurogenesis differentially contribute to the generation of PyN projection types in the neocortex (Huilgol et al, 2023). These findings suggest that indirect neurogenesis not only amplifies neuronal output but also fundamentally increases neuronal diversity (Huilgol et al, 2023; Shen et al, 2024). The complex temporal and spatial dynamics of neuron production through intermediate progenitors and their progeny create heterogeneous laminar fates.

Although it has been shown that a small fraction of IPs (IPPs) can undergo more than one round of division in rodents (Noctor et al, 2004; Postel et al, 2019; Wong et al, 2015), how IPPs contribute to neuronal number and diversity is unknown. Both the in vivo Rv-FlashTag method and in vitro time-lapse imaging show that 11–14% of RGP divisions produce IPPs. Strikingly, IPPs were mainly produced during early neurogenesis of RGPs, and contribute ~30% of PyNs in the neocortex. Our results indicate that the neuronal output of IPPs in the developing mouse neocortex is much greater than previously anticipated, and leads to systematic variabilities in neuronal number and laminar fates.

The current consensus in the field strongly supports that indirect neurogenesis represents the predominant pathway for cortical PyN generation in mammals (Cárdenas et al, 2018; Kowalczyk et al, 2009). Our clonal analysis indicates that IPs and IPPs together contribute ~64.4% of neocortical PyNs. This is somewhat lower than the estimates reported in prior studies—for instance, Vasistha et al reported 67.5% using Tbr2-CreER lines, and Huilgol et al reported 78.2% using Tbr2-2A-Flp combined with intersection/subtraction-based genetic labeling (Huilgol et al, 2023; Vasistha et al, 2015). A critical factor underlying these discrepancies may lie in the specificity of Tbr2 as a marker for IPs. Previous study indicated that Tbr2 expression extends beyond IPs to include postmitotic immature neurons, indicating broader neurogenesis regulation rather than exclusive IP identity (Englund et al, 2005). This limitation is further highlighted by comparative studies showing that in possums (mammals with a six-layered cortical structure), Tbr2$^+$ cells correspond directly to immature neurons rather than forming transient IP populations (Puzzolo and Mallamaci, 2010). These findings suggest that Tbr2-based lineage tracing may inadvertently label neurons generated through direct neurogenesis that transiently express Tbr2 during differentiation, thereby inflating the apparent contribution of indirect neurogenesis. The discrepancy between our results and those suggesting minimal direct neurogenesis may therefore reflect methodological limitations rather than genuine biological differences. Future studies must integrate complementary approaches, including single-cell lineage tracing, time-lapse microscopy, and Tbr2-independent markers, to definitively resolve the relative contributions of direct versus indirect neurogenesis and establish whether the current consensus truly reflects the underlying biology or systematic experimental bias.

Although PyN lineages showed high heterogeneity, direct and indirect neurogenesis of RGPs exhibited remarkable organization at the population level. For example, we observed that N, IP, and IPP divisions of RGPs tended to generate similar proportions of neocortical PyNs. In the first generation of RGP neurogenesis, RGPs randomly generated neurons, IPs, and IPPs. In subsequent generations, 25.9-50.0% of RGPs switched division patterns, suggesting that dynamic RGP state transitions are relatively active. Strikingly, we observed that the transition probabilities of the division patterns of RGPs remained relatively constant between generations. More importantly, our mathematical model, based on our experimental data at the population level, effectively reproduced the complex PyN lineages of the neocortex and their basic characteristics. These results strongly indicate that direct and indirect neurogenesis of RGPs are finely regulated during development. Both intrinsic and extrinsic regulatory mechanisms may govern RGP behaviors during the cortical development (Nelson et al, 2013; Nowakowski et al, 2013; Ramos et al, 2020; Vitali et al, 2018; Yoon et al, 2008). For instance, attenuation of WNT signaling has been shown to bias RGPs toward indirect neurogenesis (Vitali et al, 2018). Furthermore, growing evidence suggests that feedback signals from postmitotic neurons and IPs regulate RGP fate decisions (Nelson et al, 2013; Parthasarathy et al, 2014; Seuntjens et al, 2009; Wang et al, 2016). However, the cellular and molecular mechanism that orchestrates the temporal coordination of RGP output particularly the dynamic balance between direct and indirect neurogenesis remains largely unclear. How heterogeneous RGP lineages contribute to organized direct and indirect neurogenesis will be an important issue to address in future studies. Ultimately, such cellular and molecular analyses at single-cell resolution have the potential to reveal the underlying pathogenic mechanisms associated with neurodevelopmental diseases (Chiang et al, 2021; Guerrini and Dobyns, 2014).

# Methods

**Reagents and tools table**

| Experimental models | | |
| --- | --- | --- |
| Ai140 mouse | The Jackson Laboratory | RRID: IMSR_JAX: 030220 |
| **Antibodies** | | |
| Rat anti-Ctip2 antibody | Abcam | AB-18465 |
| Mouse anti-Satb2 | Abcam | AB-51502 |
| Rabbit anti-Pax6 | MBL | PD022 |
| Rabbit anti-PH3 | Millipore | 06-570 |
| Goat anti-rat IgG (H + L) 594 | Thermo Fisher Scientific | A-11007 |
| Donkey anti-rabbit IgG (H + L) 546 | Thermo Fisher Scientific | A-10040 |
| Goat anti-mouse IgG (H + L) 405 | Thermo Fisher Scientific | A-31553 |
| Donkey anti-rabbit IgG (H + L) 568 | Thermo Fisher Scientific | A-10042 |
| Donkey anti-mouse IgG (H + L) 647 | Molecular Probes | A-31571 |
| Donkey anti-rat IgG (H + L) 488 | Molecular Probes | A-21208 |
| **Chemicals, enzymes and other reagents** | | |
| CellTrace™ far red CFSE | Thermo Fisher Scientific | C-34564 |

| Experimental models | |
|---|---|
| **Software** | |
| GraphPad Prism v.8.0.2 | https://www.graphpad.com/ |
| Imaris v.9.0.1 | https://imaris.oxinst.com/ |
| R v.4.3.1 | https://www.r-project.org/ |
| **Other** | |
| VS120 Slide Scanning System | Olympus |
| FV3000 Confocal Laser Scanning Microscope | Olympus |
| Spin-SR Spinning Disk Microscope | Olympus |
| Andor Dragonfly Spinning Disk Microscope | Andor |

## Animals

The transgenic mouse line Ai140 (JAX #030220) was purchased from the Jackson Laboratory (Bar Harbor, ME). CD-1 mice were obtained from the Shanghai Laboratory Animal Center (SLAC). Genotyping followed the standard PCR process, and primer sequences were obtained from the Jackson Laboratory website. We regarded the day of sperm plug detection as Embryonic day 0 (E0) and the day of birth as postnatal day 0 (P0). Animals were maintained under a 12-hr light/dark cycle. All experimental procedures were conducted following the animal care guidelines at Fudan University, China.

## Immunohistochemistry

All animals were anesthetized before cardiac perfusion with PBS. Then the brains were post-fixed in 4% paraformaldehyde (PFA) at 4 °C overnight and rinsed with PBS four times, each for 5 min at room temperature. P7 mice brains in lineage tracing experiments were sequentially sectioned at a thickness of 80 μm using a Leica VT100s vibratome. Brain sections were treated with 0.5% PBST (0.5% Triton-X-100 in PBS) for 30 min and then incubated in a 10% blocking solution for 2 h at room temperature. Following this, brain sections were exposed to primary antibodies for 48 h at 4 °C and subsequently rinsed with 0.5% PBST four times at room temperature for an hour each. The brain sections were then embedded with fluorescent secondary antibodies overnight at 4 °C. Finally, coronal sections were washed with PBS three times for 30 min at room temperature.

## Antibodies

The following primary antibodies were used: rat anti-Ctip2 (1:300, Abcam, AB-18465, RRID: AB_2064130), mouse anti-Satb2 (1:300, Abcam, AB-51502, RRID: AB_882455), rabbit anti-Pax6 (1:500, MBL, PD022, RRID: AB_1520876) and rabbit anti-PH3 (1:500, Millipore, 06-570, RRID: AB_310177). Rabbit anti-Fog2 (1:200) was customized by Abcam. The following secondary antibodies were used: goat anti-rat IgG (H + L) 594 (1:250, Thermo Fisher Scientific, A-11007, RRID: AB_10561522), donkey anti-rabbit IgG (H + L) 546 (1:250, Thermo Fisher Scientific, A-10040, RRID:

AB_2534016), goat anti-mouse IgG (H + L) 405 (1:250, Thermo Fisher Scientific, A-31553, RRID: AB_221604), donkey anti-rabbit IgG (H + L) 568 (1:250, Thermo Fisher Scientific, A-10042, RRID: AB_2534017), donkey anti-mouse IgG (H + L) 647 (1:250, Molecular Probes, A-31571, RRID: AB_162542) and donkey anti-rat IgG (H + L) 488 (1:250, Molecular Probes, A-21208, RRID: AB_141709). Brain sections were immunostained with Satb2, Ctip2, and Fog2 antibodies to identify four pyramidal neuron subtypes: CPN, HPN, SCPN and CThPN.

## In utero injection

The retrovirus expressing cre (RV-cre) was derived from a stably transfected packaging cell line (293gp NIT–GFP). Carboxyfluorescein succinimidyl ester (CellTrace™ far red CFSE, Thermo Fisher Scientific, C-34564) was used at a concentration of 10 mM in DMSO. Uterine horns of pregnant mice at the E12 gestation stage were carefully exposed to a sterile environment. For sparse labeling, RV-cre (approximately 0.5 μL) was mixed with fast green (2.5 mg/mL, Sigma-Aldrich) and sterilized PBS to a low titer and injected into the embryonic cerebral ventricle using a calibrated glass micropipette with a beveled tip (Drummond Scientific, 5-000-1001-X10). After 15 to 30 min, 0.5 μL of CFSE was injected into the ipsilateral ventricle. The uterine horns were then repositioned and the peritoneal cavity was washed with approximately 10 mL of warm PBS (pH 7.4) containing antibiotics. Then the incision was closed. The pregnant mice were placed on a warm pad until they recovered and were subsequently returned to their standard housing conditions.

## Imaging and image processing

In the lineage tracing experiment, brain slices were scanned for five fluorescent signals, including 405 (Satb2), 488 (EGFP), 546 (Fog2), 594 (Ctip2) and 647 (CFSE). To identify EGFP and CFSE coexpression lineages, complete sections were systematically scanned, rostrally to caudally, utilizing a 20× objective lens on a fluorescence microscope (Olympus, VS120). Pre-selected lineages that exhibited coexpression of EGFP and CFSE were subjected to further scanning using a 60× oil objective lens on a spinning-disk confocal microscope (Oxford Instruments, Andor Dragonfly) equipped with an Electron-Multiplying CCD (EMCCD) to enhance the capture of far red signals. The step size was 0.33 μm, as recommended by the manufacturer. The fluorescence intensities of clonal neurons were at least twice that of the background. During imaging, we ensured that the neuron with the highest fluorescence intensity in an individual lineage was not overexposed. The same scanning parameters were applied to an individual lineage. To classify PyN types, Satb2, Fog2, and Ctip2 signals were scanned with 405, 561, and 594 lasers, and a virtual channel module was used to achieve display of the five channels simultaneously. The CFSE fluorescence intensities of the cell bodies were measured using the surface module of IMARIS (10.1.1). 3D-box in IMARIS was used to frame the neuron soma and then the sum of fluorescence intensity values of the soma within the 3D-box was measured. Automatic background subtraction was used during fluorescence intensity measurement. The parameters, including thresholding (background subtraction) and smooth (surface detail), were kept the same in individual lineages during image processing.

## Cortical slice preparation and time-lapse imaging

In the time-lapse lineage tracing experiment, following the injection of RV-cre into E12 Ai140 embryonic ventricles, embryos were harvested at E13. In the CFSE fluorescence intensity tracing experiment, CFSE was injected into E12 Ai140 embryonic ventricles and embryo brains were obtained at E13. 200 μm thick cortical slices were carefully sectioned using a vibratome (Leica, VT100s) in chilled artificial cerebrospinal fluid (ACSF) containing the following components (in mM): 125 NaCl, 2.5 KCl, 1 MgCl$_2$, 2 CaCl$_2$, 1.25 NaH$_2$PO$_4$, 25 NaHCO$_3$, and 25 D-(+)-glucose, continuously oxygenated with 95% O$_2$ and 5% CO$_2$. The coronal vibratome sections were then transferred to a glass-bottom dish (MatTek) immersed in a specialized cortical slice culture medium comprising 66% BME, 25% Hanks, 5% FBS, 1% N-2, 1% penicillin/streptomycin, glutamine (all from Invitrogen), and 0.66% D-(+)-glucose (Sigma-Aldrich). For time-lapse imaging, the culture was relocated to an inverted Olympus Spin-SR confocal microscope, equipped with an on-stage incubator (Okolab), which maintained a continuous flow of 5% CO$_2$, 5% O$_2$, and balanced N$_2$, while sustaining a temperature of 37 °C. Slices were imaged at 30-minute intervals for a duration of up to 72 h (E13-E16) with thick z-stacks. Maximum intensity projections of the acquired stacks were analyzed using IMARIS (10.1.1) software. Processed image series were subsequently transformed into video using Python (version 3.12.0).

## Lineage tree reconstruction

Our lineage tree reconstructions were based on automated reconstruction followed by manual verification.

### Automatic reconstruction

To construct a lineage tree based on a group of fluorescence intensity values, we first denoted the highest fluorescence intensity value as $\varphi_1$ (Fig. EV4A). Then we set up a series of fluorescence intensity "hierarchy". The first hierarchy had the highest fluorescence intensity which was denoted as $\eta_1$ ($\eta_1 = \varphi_1$). The second hierarchy $\eta_2$ had half fluorescence intensity of $\varphi_1$ ($\eta_2 = \frac{1}{2}\varphi_1$). The third hierarchy had half fluorescence intensity of $\eta_2$ ($\eta_3 = \frac{1}{2}\eta_2 = \frac{1}{2^2}\varphi_1$), and so forth. The assignment of remaining neurons to different intensity hierarchies was based on the proximity of their fluorescence intensities to these specified values ($\eta_1, \eta_2, \eta_3...\eta_n$). Thus, $\eta_n$ had the lowest fluorescence intensity and $\eta_1$ had the highest (Fig. EV4B).

Then we performed binary tree generation following three rules. Rule 1 performs permutation ($A_n^n = n!$) from the lowest hierarchy $\eta_n$ to the highest hierarchy $\eta_1$. The lowest hierarchy has the lowest fluorescence intensity and has only neurons involved, while higher hierarchies which have generated new nodes according to Rule 3 have both neurons and progenitors involved in the permutation. For example, in $\eta_4$, two neurons participate in the permutation while in $\eta_3$ there are two neurons and a progenitor, and in $\eta_1$, there is one neuron and three progenitors (Fig. EV4C). Based on Rule 1, in every individual permutation, Rule 2 combines the first two adjacent cells as a group and then the next two cells as another group when there are four neurons in the hierarchy. If there are only three cells in the hierarchy, the first two cells are combined as a group and the remaining individual cell is regarded as a group

(Fig. EV4C). Then based on Rule 2, in Rule 3, combined groups will generate new nodes (new cells, progenitors) in the higher hierarchies according to 9 permitted combinations and 7 prohibited combinations. IPP, IP, and RGP are denoted as $\alpha$, $\beta$ and $\gamma$, respectively. $\kappa$ stands for neurons. In the nine permitted combinations, a combination of two cells can generate new nodes in the next hierarchy, whereas in the seven prohibited combinations, they cannot generate new nodes (Fig. EV4C). The seven prohibited combinations include $\kappa\beta, \beta\kappa, \kappa\alpha, \alpha\kappa, \beta\alpha, \alpha\beta, \alpha\alpha$ and the nine permitted combinations are as follows:

$$[\beta\gamma, \gamma\beta, \kappa\gamma, \gamma\kappa, \alpha\gamma, \gamma\alpha, \gamma\gamma] = \gamma$$

$$[\kappa\kappa] = \beta$$

$$[\beta\beta] = \alpha$$

Then we performed recursion of the three Rules from the lowest hierarchy to the highest hierarchy (Fig. EV4D). After the end of the recursion, the lineage tree is formed which is saved as a binary tree (Fig. EV4E). Finally, to eliminate duplications, cell left-right exchanges and tree branch left-right exchanges were removed (Fig. EV4F). Visualization of lineage trees was accomplished through the Python Tkinter and all operations were programmed using Python (3.12.0).

### Manual evaluation

Based on the machine construction, we further proofread and evaluated the lineages manually. After duplication elimination, automated reconstruction provided several options for lineage trees. Three neurons with similar fluorescence intensities will result in multiple lineage trees. Therefore, when determining which two of the three neurons with similar fluorescence intensities were derived from the same IP, we followed the rule that neurons originating from the same IP were positioned in closer proximity since sister neurons from the same IP tend to be located much closer together than other neurons (Mihalas and Hevner, 2018; Qian et al, 1998). Likewise, four neurons with similar fluorescence intensities derived from an IPP will also result in multiple lineage trees. Therefore, when deciding which two of the four cells with similar fluorescence intensities originated from the same IP, we determined that the two neurons in closer proximity were likelier to have originated from the same IP.

## Reconstruction of in silico lineages

In asymmetric RGP lineages, $RGP_n$ meant this RGP divided "n" times and it was the "n"th generation. $RGP_n$ can divide and give rise to a new RGP named $RGP_{n+1}$ and a progeny which can be IPP, IP or neuron. Progeny of $RGP_n$ was denoted as $C_{n+1}$ and $C_{n+1} = \{IPP, IP, N\}$.

Our model was based on four rules:

Rule 1: The number of neurons that can be produced by an RGP represents the proliferative capacity of the RGP, which is defined as PyN. We first found that the PyN distribution of experimental lineages matched mixed Poisson distribution and performed a chi-square test to validate this. The P value was 0.2421. It could be considered that the distribution of lineage size conforms to the

mixture distribution below:

$$PyN = 0.35 \times t - Poisson(\lambda_1 = 4.09) + 0.65 \times t - Poisson(\lambda_2 = 7.62)$$

Rule 2: We defined the probabilities of the initial RGP generating IPP, IP, and neuron (N) as being equal which means that the RGP randomly chooses division patterns at the beginning as indicated in the experimental lineages.

$$P\{C_1 = IPP\} = P\{C_1 = IP\} = P\{C_1 = N\} = \frac{1}{3}$$

Rule 3: We defined the transition probabilities of RGP division patterns between generations as being stable. We first validated this rule by performing an independence test on the number of different division patterns of $RGP_n$ and $RGP_{n+1}$. The P value is less than $10^{-6}$, proving that the division pattern of the previous generation of RGP is dependent on the division pattern of the next generation. This indicated a correlation of division pattern transition probabilities between RGP generations.

The transition probabilities are defined as $p_{ij}$ in which $i$ stands for RGP division pattern of the former generation and $j$ stands for division pattern of the latter generation. $i$ and $j$ can be 1, 2 or 3. 1 stands for IPP, 2 stands for IP and 3 stands for neuron. P is a set of many $p_{ij}$ values and can be written as a matrix:

$$P = \begin{bmatrix} p_{11} & p_{12} & p_{13} \\ p_{21} & p_{22} & p_{23} \\ p_{31} & p_{32} & p_{33} \end{bmatrix} \text{ in which } \sum_{j=1}^{3} p_{ij} = 1 (i = 1, 2, 3)$$

In our tests, except for the fluctuation of $p_{1j}$, the other 6 transition probabilities $p_{2j}$ and $p_{3j}$ are all stable and independent of generations. A small number of IPP division cases may result in the fluctuation of $p_{1j}$.

Rule 4: When the clone size or the remaining number of neurons ($PyN_{remain}$) after sequential divisions is less than 4, then no more IPPs can be generated ($2 \leq PyN_{remain} < 4$) or no more IPs or neurons can be generated ($PyN_{remain} = 1$). At the same time, the transition probability matrix P in rule 3 also changes accordingly.

$$P = \begin{bmatrix} 0 & \frac{p_{12}}{p_{12}+p_{13}} & \frac{p_{13}}{p_{12}+p_{13}} \\ 0 & \frac{p_{22}}{p_{22}+p_{23}} & \frac{p_{23}}{p_{22}+p_{23}} \\ 0 & \frac{p_{32}}{p_{32}+p_{33}} & \frac{p_{33}}{p_{32}+p_{33}} \end{bmatrix}, (2 \leq PyN < 4)$$

$$P = \begin{bmatrix} 0 & 0 & 1 \\ 0 & 0 & 1 \\ 0 & 0 & 1 \end{bmatrix}, (PyN = 1)$$

Based on the above four rules, as long as $\lambda_1$, $\lambda_2$ and $p_{ij}$ are given, lineages can be simulated.

## Quantification of cell distributions

The relative positions of neurons in the neocortex were defined as the distance of a neuron from the pia mater divided by the distance from the pia mater to the subplate. These distances were quantified using Neurolucida (MBF Bioscience) software. The layers where neurons were located were determined using the Allen Brain Atlas (https://mouse.brain-map.org/experiment/thumbnails/100048576?image_type=atlas).

## Statistical analysis

In the cumulative distance analysis, we first separated neurons in a lineage into two groups: one group was comprised of neurons with the same fluorescence intensity (denoted as M) and the other group was comprised of the remaining neurons in the lineage (denoted as N). We then measured the distance between the M group neurons. We also measured the distance between the neurons of N as well as distance between the neurons of M and N. We then made two cumulative curves in which the shorter cumulative distance reflects clustering, whereas the longer cumulative distance reflects dispersal.

For all statistical analyses in this study, one-way ANOVA was used when dealing with data involving more than two groups and Student's $t$ test was used for data with two groups. For datasets did not pass Shapiro–Wilk test, Kruskal–Wallis one-way ANOVA, paired Student's $t$ test followed by Wilcoxon matched-pairs signed rank test, and unpaired Student's $t$ test followed by Mann–Whitney test was applied, otherwise, one-way ANOVA was applied. Kullback–Leibler Divergence analyses were used in measuring the degree of difference between distributions of experimental lineages and in silico lineages (Fig. 5G–J). All statistical results were presented as mean ± s.e.m. unless otherwise stated in the legends. Curve fitting was executed through the curve fitting function in MATLAB (R2016a).

## Data availability

The computer codes produced for lineage tree reconstruction and lineage simulation are available in the following database: https://github.com/dan20231213/simulated_lineages.git and https://github.com/dan20231213/lineage_tree_construction.git. The source data of this paper are collected in the following database record: Biostudies: https://www.ebi.ac.uk/biostudies/bioimages/studies/S-BIAD2154.

The source data of this paper are collected in the following database record: biostudies:S-SCDT-10_1038-S44318-025-00624-9.

## Peer review information

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

## Acknowledgements

This work was supported by the National Key Research and Development Program of China (2021ZD0202500), the Natural Science Foundation of China (31930044, 31725012), the Foundation of Shanghai Municipal Education Commission (2019-01 07-00-07-E00062), the Collaborative Innovation Program of Shanghai Municipal Health Commission (2020CXJQ01), the Shanghai Municipal Science and Technology Major Project (No.2018SHZDZX01) and ZJLab to Y-CY The National Key Research and Development Project of China (Grant No. 2023YFF 1204802 to YZ), STI2030 Major Projects (Grant No. 2021ZD0200100 to YZ), the National Natural Science Foundation of China (Grant No. 82071259 to YZ). This work was the result of using research equipment shared in the Institute of Brain Science, Fudan University.

## Author contributions

**Dan Shen**: Conceptualization; Data curation; Formal analysis; Validation; Investigation; Methodology; Writing—original draft. **Xin-Yi Wang**: Investigation. **Ruo-Hang Liu**: Formal analysis; Investigation. **Huan-Huan Deng**: Investigation. **Shi-Yuan Tong**: Data curation. **Jun-Yang Chen**: Investigation. **Zi-Yun Zhai**: Resources. **Yuan-Xin Li**: Resources. **You-Ning Lin**: Investigation. **Fu-Wei Yang**: Investigation. **Chen-Xi Wang**: Investigation. **Lin-Yun Liu**: Conceptualization; Supervision; Writing—review and editing. **Ying Zhu**: Formal analysis; Investigation. **Yong-Chun Yu**: Conceptualization; Resources; Supervision; Funding acquisition; Project administration; Writing—review and editing.

Source data underlying figure panels in this paper may have individual authorship assigned. Where available, figure panel/source data authorship is listed in the following database record: biostudies:S-SCDT-10_1038-S44318-025-00624-9.

## Disclosure and competing interests statement

The authors declare no competing interests.

# Expanded View Figures

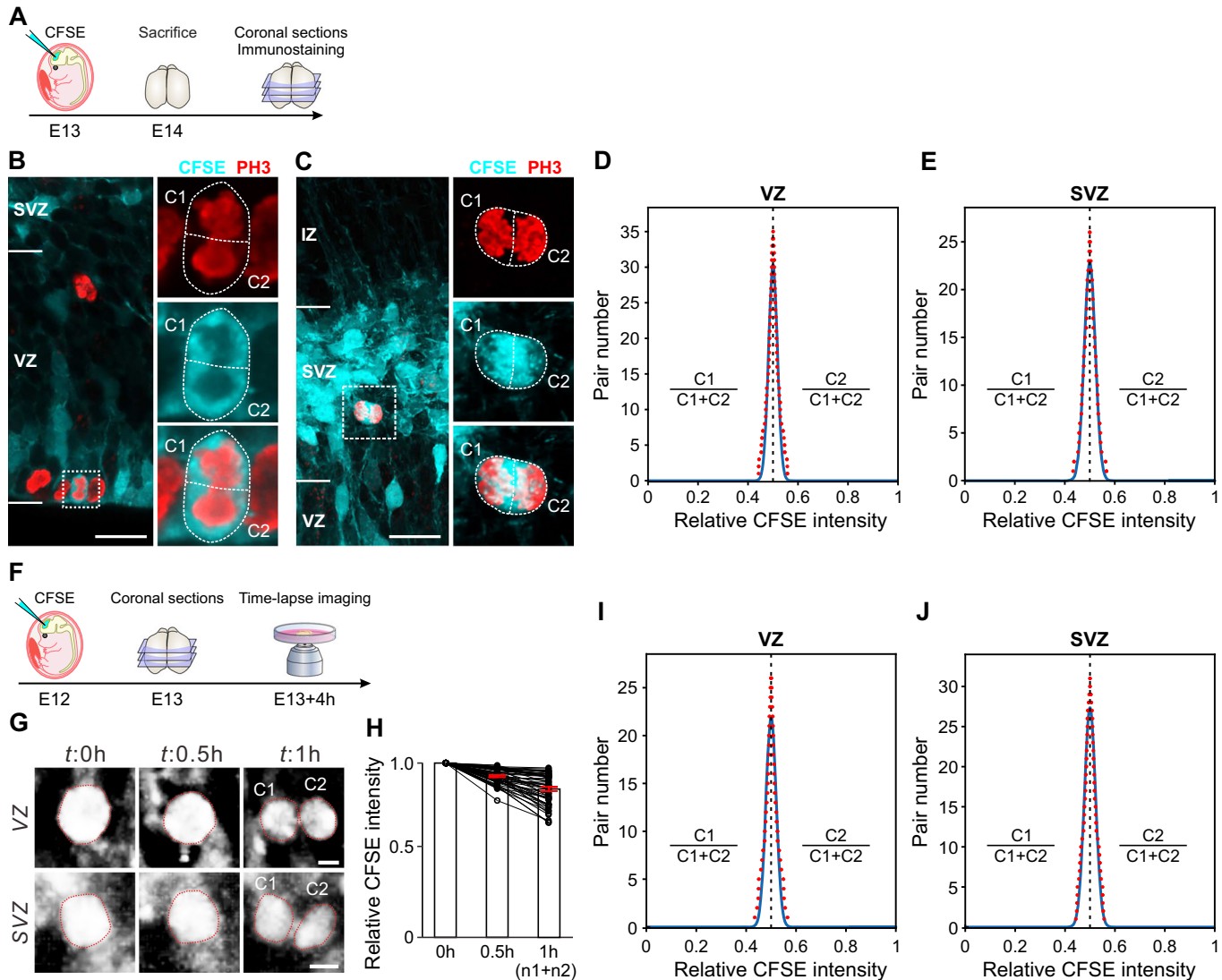

**Figure EV1. CFSE fluorescence intensities of progenitors were halved in daughter pairs in the embryonic neocortex.**

(**A**) Experimental paradigm of labeling dividing progenitors with CFSE. (**B**) Images of a CFSE/PH3-positive dividing progenitor cell in the VZ of E14 neocortex. C1: cell 1, C2: cell 2. Scale bar: 20 µm. (**C**) Image of a CFSE/PH3-positive dividing progenitor in the SVZ of E14 neocortex. C1: cell 1, C2: cell 2. Scale bar: 20 µm. (**D, E**) Quantification of relative CFSE fluorescence intensities for sister pairs in the VZ (D) and SVZ (E), fitted with Gaussian curves. (**F**) Experimental paradigm of time-lapse imaging of CFSE progenitors. (**G**) Time-lapse imaging of CFSE-labeled dividing progenitors in the VZ and SVZ. C1 and C2 represent daughter pairs generated by an RGP. Scale bar: 5 µm. (**H**) Quantification of relative CFSE fluorescence intensities of dividing progenitors and sister pairs, showing a decrease in fluorescence intensities over the course of imaging ($n = 57$). Data are shown as mean ± SEM. (**I, J**) Quantification of relative CFSE fluorescence intensities for sister pairs in the VZ (I) and SVZ (J) in time-lapse experiments, fitted with Gaussian curves.

 

**A**

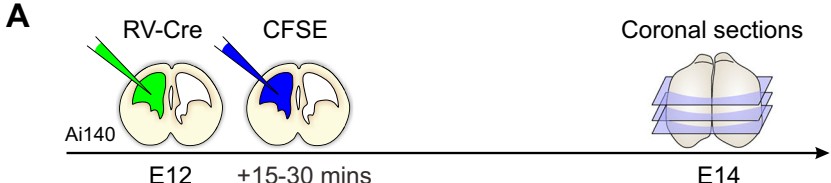

RV-Cre   CFSE                    Coronal sections

Ai140

E12    +15-30 mins                    E14

**B**

| GFP | GFP+GFP surface | GFP surface | GFP in surface | CFSE in surface | GFP+CFSE in surface |
|-----|------|------|------|------|------|

Daughter (repeated across panels)

RGP (repeated across panels)

**C**

Pax6 | Pax6 + GFP+CFSE in surface

Daughter

RGP

**D**

CFSE intensity

49.3×10⁶

14.6%
8.8×10⁶

100.0%
60.4×10⁶

Daughter

85.4%
51.6×10⁶

RGP

**E**

Percentage (%)

RGP    RGP process

9.8%

**F**

Percentage (%)

ns

RGP soma    Daughter

**Figure EV2.  Sparse distribution of CFSE fluorescence signals in RGP processes.**

(A) Experimental paradigm for labeling RGPs and their processes using Rv-FlashTag. (B) Confocal images of a lineage, depicting the soma and process of an RGP and its first-generation daughter cell, both labeled with RV-cre and CFSE. The cells were outlined using Surface function in Imaris software. Scale bar: 15 μm. (C) Immunohistochemical staining for Pax6 in lineage. Scale bar: 15 μm. (D) CFSE fluorescence intensity measurements for RGP soma, RGP process and daughter cell. Black numbers indicate CFSE fluorescence intensity and red numbers indicate the percentages of CFSE fluorescence in RGP soma, process, and the entire sample. (E) Proportions of CFSE fluorescence intensities in RGP processes ($n = 57$). Data are shown as mean ± SEM. (F) Relative fluorescence intensity comparison between RGP somas and daughter cells ($n = 57$). Data are shown as mean ± SEM. $P = 0.2783$ (paired Student's $t$ test followed by Wilcoxon matched-pairs signed rank test).

                                                                        

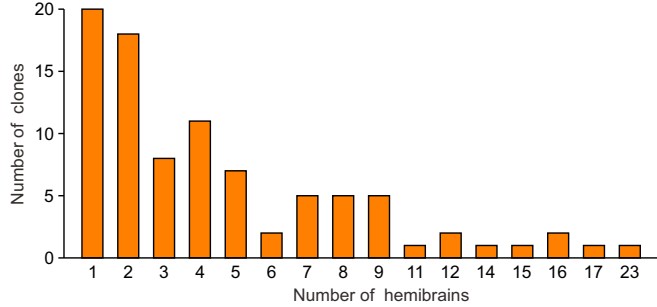

**Figure EV3. Clones were sparsely labeled in the neocortex by Rv-FlashTag.**

Quantification of the number of Rv-FlashTag labeled clones in individual hemibrains.

**Figure EV4. Automated lineage tree reconstruction.**

(A) Input data of CFSE fluorescence intensity values of lineages. (B) Hierarchical organization of neurons based on varying CFSE intensities. (C) Three rules of binary tree generation: permutation, pair combination, and new node generation. (D) Recursive application of the three rules from the lowest to the highest CFSE intensity hierarchy. (E) Sample binary tree generation. (F) Removal of duplicated lineages.

## A | Gain bias from experimental lineages

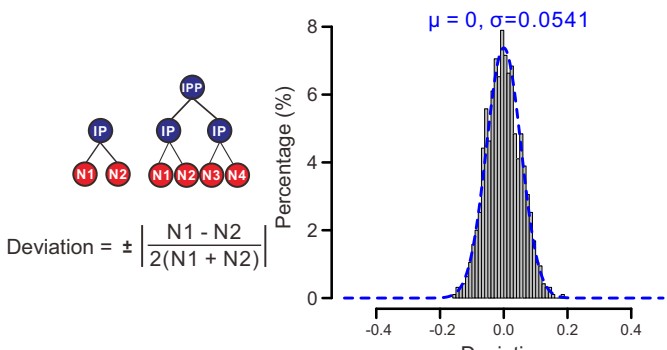

$$\text{Deviation} = \pm \left| \frac{N1 - N2}{2(N1 + N2)} \right|$$

μ = 0, σ=0.0541

## B | Generate in silico lineages

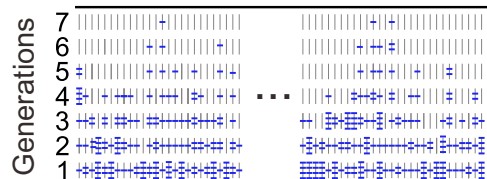

**100 lineages**

Generations 1–7 shown vertically.

**Division patterns**
+ Direct neurogenesis
‡ IP-mediated indirect neurogenesis
‡ IPP-mediated indirect neurogenesis

## C | Assign values to in silico lineages

**Standard lineages**

```
<key>Y001</key>
<string>50.00,25.00,12.50,1.56,1.56,1.56,1.56,1.56</string>
<key>Y002</key>
<string>25.00,25.00,25.00,12.50,6.25</string>
<key>Y003</key>
<string>50.00,12.50,12.50,6.25,6.25</string>
<key>Y004</key>
<string>12.50,12.50,12.50,12.50,6.25,6.25,6.25,6.25</string>
<key>Y005</key>
<string>25.00,25.00,12.50,12.50,6.25,6.25,6.25</string>
<key>Y006</key>
<string>25.00,12.50,12.50,12.50,12.50</string>
<key>Y007</key>
<string>50.00,6.25,6.25,6.25,6.25,6.25,6.25,6.25</string>
<key>Y008</key>
<string>50.00,12.50,12.50,6.25,6.25,6.25</string>
```

## D | Perturbation to standard lineages

**Perturbed lineages**

```
<key>Y001</key>
<string>45.53,25.67,12.74,1.65,1.64,1.54,1.53,1.52,1.51</string>
<key>Y002</key>
<string>25.86,23.64,23.39,10.98,6.45</string>
<key>Y003</key>
<string>45.54,12.90,12.13,6.28,6.21</string>
<key>Y004</key>
<string>14.46,12.88,12.62,11.73,6.43,6.06,5.92,5.83</string>
<key>Y005</key>
<string>26.09,24.89,13.40,13.01,6.63,6.08,5.95</string>
<key>Y006</key>
<string>25.83,13.48,13.09,12.72,11.54</string>
<key>Y007</key>
<string>54.69,6.61,6.29,6.17,6.05,5.78,5.66,5.59</string>
<key>Y008</key>
<string>51.77,13.12,12.70,7.06,7.00,6.72</string>
```

Disturbed CFSE.intensity = standard CFSE.intensity x (1 + δ)

δ: Diviation values that randomly sampled from the normal distribution based on probabilities.

## E | Automated lineage tree reconstruction of unperturbed lineages and perturbed lineages

Unperturbed lineages | Perturbed lineages

Unperturb.001 / Unperturb.002 (left), Perturb.001 / Perturb.002 (right)

● RGP
● IPP
● IP
● N

## F | Assess whether lineage trees were altered following perturbation

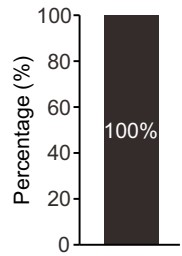

100%

■ lineage tree unchanged

**Figure EV5.  Validation of CFSE distribution deviations on lineage reconstruction accuracy.**

(**A**) CFSE fluorescence deviations between daughter cells follow a normal distribution with mean (μ) = 0, standard deviation (σ) = 0.0541, and 99% confidence interval = ±0.139. The deviation is calculated as ±|(N1 - N2)/2(N1 + N2)|, where N1 and N2 represent CFSE intensities in daughter cells. (**B**) 100 in silico lineage trees with different division patterns. Generations 1-7 are shown vertically. (**C**) Control dataset: unperturbed CFSE intensity values for representative in lineage trees, formatted in XML with unique identifiers. Initial CFSE fluorescence intensity is normalized to 100 for the founding cell. (**D**) Test dataset: lineage data after applying experimentally-derived fluorescence perturbations. Perturbed CFSE intensity = standard intensity × (1 + δ), where δ values are randomly sampled from the experimentally observed normal distribution. (**E**) Automated lineage reconstruction comparison demonstrates identical tree topologies for unperturbed (left) versus perturbed (right) datasets. (**F**) Quantitative assessment of reconstruction accuracy across all 100 simulated lineage trees. Bar graph shows 100% of lineage trees remained unchanged following perturbation.

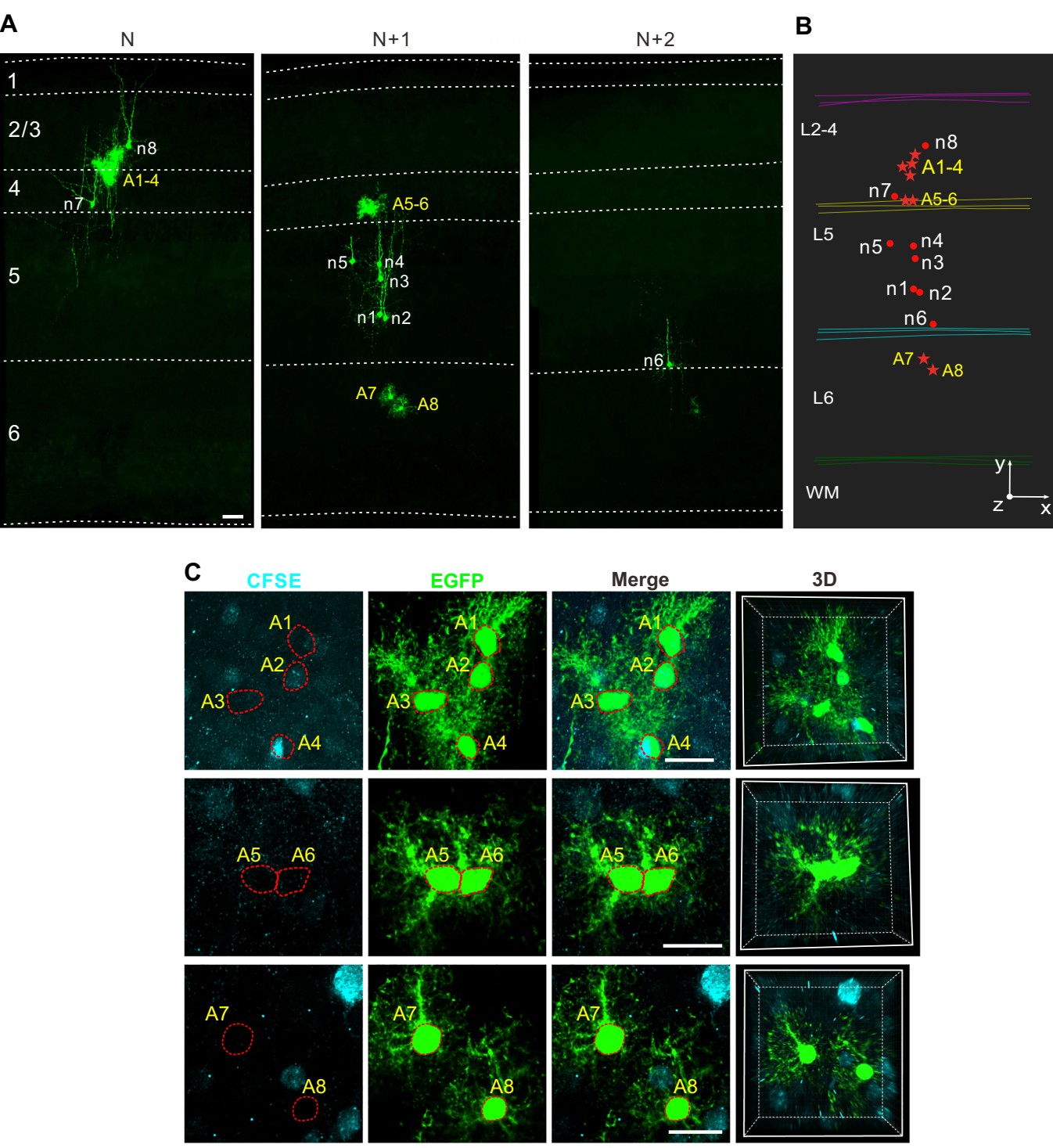

**Figure EV6.  Rv-FlashTag labeled glial lineages.**

(**A**) Confocal images of an Rv-FlashTag labeled clone that contains eight sister neurons (n1–n8) and eight glial cells (A1–A8). Scale bar: 50 μm. (**B**) 3D reconstruction of the clone in (**A**). Red dots reprsent neurons and red stars represent astrocytes. WM white matter. (**C**) High-magnification images of glial cells in this clone. Except for A2, other glial cells were unlabeled with CFSE. Scale bar: 20 μm.

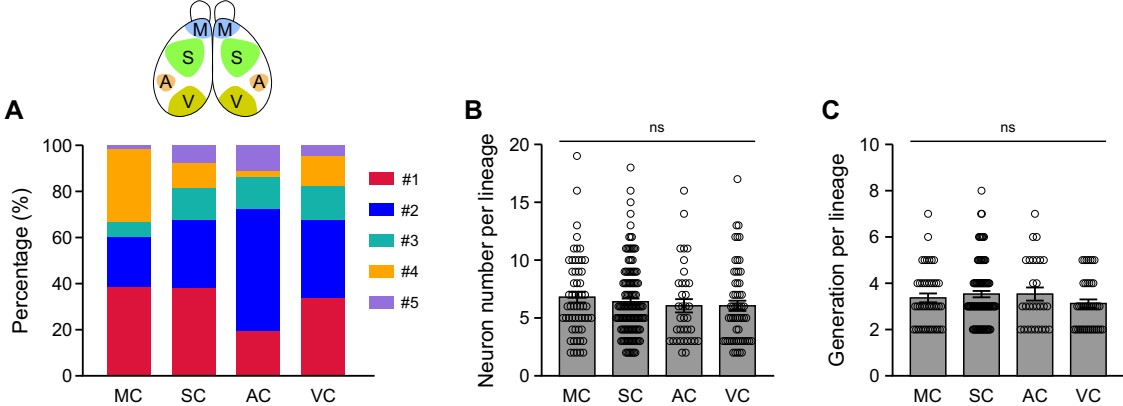

Figure EV7. **Characteristics of lineages across cortical regions.**

(A) Proportions of lineage clusters in motor cortex (MC), somatosensory cortex (SC), auditory cortex (AC), and visual cortex (VC). The schematic diagram above illustrates the relative positions of four cortical regions in the brain. (B) Clone sizes in MC ($n = 56$), SC ($n = 130$), AC ($n = 36$) and VC ($n = 58$). Data are shown as mean ± SEM. $P = 0.4477$ (Kruskal–Wallis one-way ANOVA). (C) Number of generations per lineage in the MC ($n = 43$), SC ($n = 97$), AC ($n = 28$) and VC ($n = 40$). Data are shown as mean ± SEM. $P = 0.4694$ (Kruskal–Wallis one-way ANOVA).

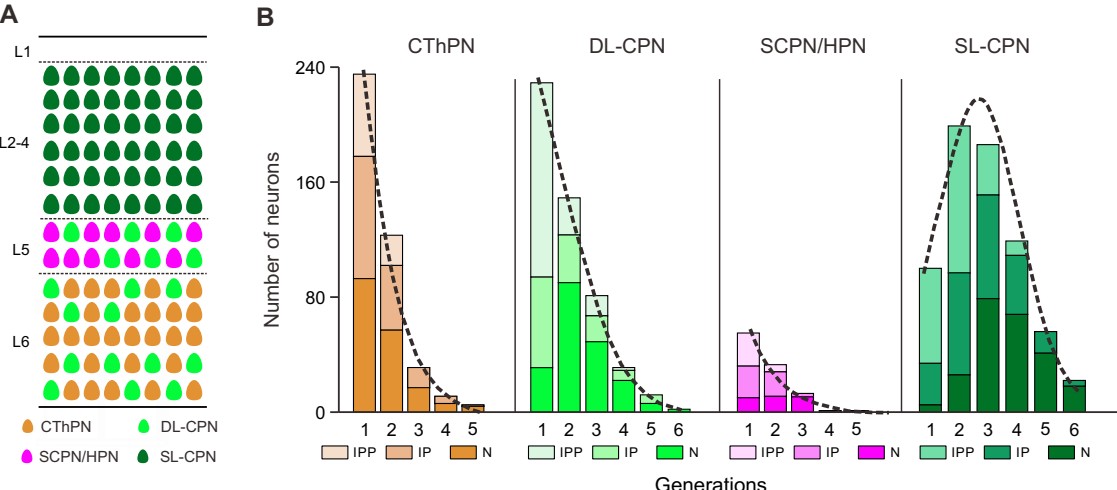

**Figure EV8.** **CThPN, DL-CPN, SCPN/HPN and SL-CPN neuronal output across generations.**

(**A**) Layer distribution of different types of pyramidal neurons. (**B**) CThPN, DL-CPN, SCPN/HPN and SL-CPN numbers generated in G1 to G5/6 in all lineages. The fitting curves of CThPN, DL-CPN and SCPN/HPN are the exponential decay. The fitting curve of SL-CPN is a Gaussian distribution.

