## [Peer Review File · The EMBO Journal]

Direct and indirect neurogenesis from radial glial progenitor cell clones in the mouse neocortex

Dan Shen, Xin-Yi Wang, Ruo-Hang Liu, Huan-Huan Deng, Shi-Yuan Tong, Jun-Yang Chen, Zi-Yun Zhai, Yuan-Xin Li, You-Ning Lin, Fu-Wei Yang, Chen-Xi Wang, Lin-Yun Liu, Ying Zhu, and Yong-Chun Yu

Corresponding author(s): Yong-Chun Yu (ycyu@fudan.edu.cn) , Lin-Yun Liu (liuly@fudan.edu.cn), Ying Zhu (ying_zhu@fudan.edu.cn)

Review Timeline:

Submission Date:	11th Feb 25
Editorial Decision:	14th Mar 25
Revision Received:	16th Jul 25
Editorial Decision:	5th Aug 25
Revision Received:	11th Sep 25
Accepted:	28th Sep 25

Editor: Ioannis Papaioannou

Transaction Report:

Dear Prof. Yu,

Thank you for submitting your manuscript EMBOJ-2025-120450 for consideration by The EMBO Journal, and for your patience during peer review. Your manuscript has now been seen by three experts in the field, and we have received the full set of their comments, which you can find below.

As you will see, the referees recognize that this is an interesting study presenting high-quality data and providing a significant advance over the literature in the field of cortical development. However, they also identify a number of limitations that would have to be addressed experimentally for strengthening the manuscript further and increasing its impact on the field. There are also some concerns regarding the depth of the discussion, which according to some of the referees' suggestions, should include some more relevant work from other groups that is missing from the current version of the manuscript, and also clarify certain ambiguities or apparent contradictions with the literature.

Given the referees' positive comments and recommendations, I would like to invite you to submit a thoroughly revised version of your manuscript along with a detailed point-by-point response addressing all referees' comments. I should add that it is The EMBO Journal policy to allow only a single round of major revision, and acceptance of your manuscript will therefore depend on the completeness of your responses in this revised version. Please let me know if you have any questions or comments that you would like to discuss with me. If there are any major points you do not agree with or cannot address during your revision, I would encourage you to share them with me as early as possible to discuss how to proceed further in the most efficient way.

We generally allow three months as standard revision time (June 13, 2025). As a matter of policy, competing manuscripts published during this period will not negatively impact our assessment of the conceptual advance presented by your study. However, we request that you contact us as soon as possible upon publication of any related work, to discuss how to proceed. Should you foresee a problem in meeting this three-month deadline, please let us know in advance and we may be able to grant an extension.

Thank you for the opportunity to consider your work for publication in The EMBO Journal. I look forward to your revision.

Best regards,

Ioannis

Instructions for preparing your revised manuscript

1. When you are ready to submit the revision, please upload:

- A Word file of the manuscript text (including legends of main Figures, EV Figures and Tables). Please make sure that changes are highlighted (or "tracked") to be clearly visible.

- Individual production-quality figure files (one file per figure). When assembling your figures, please refer to our figure preparation guidelines in order to ensure proper formatting and readability in print as well as on screen:

If the data shown in a figure are obtained from n {less than or equal to} 2, please use scatter plots showing the individual data points.

- i. the name of the statistical test used to generate error bars and P values
- ii. the number (n) of independent experiments (please specify technical or biological replicates) underlying each data point (discussion of statistical methodology can be reported in the Materials and Methods section, but figure legends should contain a basic description of n , P, and the test applied)
- iii. the nature of the bars and error bars (s.d., s.e.m.).

- A point-by-point response to the referees' comments, with a detailed description of the changes made (as a word file). All referees' concerns must be fully addressed and their suggestions taken on board. When preparing your letter of response to the referees' comments, please bear in mind that this will form part of the Review Process File and will therefore be available online to the community. Please note that you have the possibility to opt out of the transparent process at any stage prior to publication by letting the editorial office know (contact@embopress.org); if you do opt out, the Review Process File link will point to the following statement: "No Review Process File is available with this article, as the authors have chosen not to make the review process public in this case.". For more details on our Transparent Editorial Process, please visit our website: <https://www.embopress.org/page/journal/14602075/authorguide#transparentprocess>

- Expanded View (EV) files (replacing Supplementary Information) that are collapsible/expandable online. A maximum of 5 EV Figures can be typeset. EV Figures should be cited as "Figure EV1, Figure EV2" etc. in the text, and their respective legends should be included in the manuscript file after the legends of regular figures. See detailed instructions regarding Expanded View files here: <https://www.embopress.org/page/journal/14602075/authorguide#expandedview>

- For the figures that you do NOT wish to display as Expanded View figures, they should be bundled together with their legends in a single PDF file called "Appendix", which should start with a short Table of Contents (including page numbers). Appendix figures should be referred to in the main text as: "Appendix Figure S1, Appendix Figure S2" etc. Please see detailed instructions here: <https://www.embopress.org/page/journal/14602075/authorguide#expandedview>

- A complete author checklist, which you can download from our author guidelines (<https://www.embopress.org/page/journal/14602075/authorguide>). Please note that the checklist will also be part of the Review Process File.

2. Please note that no statistics should be calculated and shown in Figures if $n=2$. Please also note that each p value should be reported as an exact value.

3. Before submitting your revision, primary datasets (and computer code, where appropriate) produced in this study need to be deposited in appropriate public databases (see <https://www.embopress.org/page/journal/14602075/authorguide#dataavailability>). The accession numbers, database, and the specific URLs (links) should be listed in a formal "Data availability" section (placed after Methods), following the example below:

"The RNA-seq datasets produced in this study are available in the following database:
Gene Expression Omnibus GSE46843 (<https://www.ncbi.nlm.nih.gov/geo/query/acc.cgi?acc=GSE46843>)"

*** All links should resolve to a page where the data can be accessed. ***

*** Please remember to provide in the Data availability section of your revised manuscript reviewer passwords if the datasets are not yet public. ***

*** The Data Availability Section is restricted to new primary data that are part of this study. In case you have no data that require deposition in a public database, please state so instead of referring to the database: "Our study includes no data deposited in public repositories." under the heading "Data availability". ***

4. The materials and methods need to be described in the manuscript using our structured methods format, which is now required for all research articles. According to this format, the Methods section includes a single "Reagents and Tools Table" - listing key reagents, experimental models, software and relevant equipment including their sources and relevant identifiers- followed by a "Methods and Protocols" section describing the methods. Please download and fill our Reagents and Tools Table template (.docx), which you can find in our author guide:

<https://www.embopress.org/page/journal/14602075/authorguide#structuredmethods>. When submitting your revised manuscript, please do not include the Reagents and Tools Table in the Methods section of the manuscript but instead upload it as a separate file choosing the file type "Reagent Table".

5. Please check that the title and the abstract of the manuscript are brief, yet explicit, even to non-specialists. The length of the title should not exceed 100 characters, and the abstract should be a single paragraph not exceeding 175 words.

6. Please also note our reference format: <https://www.embopress.org/page/journal/14602075/authorguide#referencesformat>.

8. Please remember: digital image enhancement is acceptable practice, as long as it accurately represents the original data and

conforms to community standards. If a figure has been subjected to significant electronic manipulation, this must be noted in the figure legend or in the "Materials and Methods" section. The editors reserve the right to request original versions of figures and the original images that were used to assemble the figure.

9. Our journal encourages inclusion of data citations in the reference list to directly cite datasets that were obtained from public databases. Data citations in the article text are distinct from normal bibliographical citations and should directly link to the database records from which the data can be accessed. In the main text, data citations are formatted as follows: "Data ref: Smith et al, 2001" or "Data ref: NCBI Sequence Read Archive PRJNA342805, 2017". In the Reference list, data citations must be labeled with "[DATASET]". A data reference must provide the database name, accession number/identifiers, and a resolvable link to the landing page from which the data can be accessed at the end of the reference. Further instructions are available at: <https://www.embopress.org/page/journal/14602075/authorguide#referencesformat>.

10. We request authors to consider both actual and perceived competing interests. Please review our policy (<https://www.embopress.org/page/journal/14602075/authorguide#conflictsofinterest>) and update your competing interests statement if necessary. Please name this section 'Disclosure and competing interests statement' and place it after the Acknowledgements section.

11. Please note that all corresponding authors are required to provide an ORCID ID upon submission of a revised manuscript (<https://orcid.org/>). Please find instructions on how to link your ORCID ID to your account in our manuscript tracking system in our Author guidelines (<https://www.embopress.org/page/journal/14602075/authorguide#authorshipguidelines>).

12. We use CRediT to specify the contributions of each author in the journal submission system. CRediT replaces the author contribution section, which should be removed from the manuscript. Please use the free text box to provide more detailed descriptions. See also guide to authors: <https://www.embopress.org/page/journal/14602075/authorguide#authorshipguidelines>.

14. We would also welcome the submission of cover suggestions or motifs to be used by our Graphics Illustrator in designing a cover.

15. Please use the link below to submit your revision:
<https://emboj.msubmit.net/cgi-bin/main.plex>

Referee #1:

In their study, the authors addressed an important question in cortical development. How can radial glial progenitors (RGPs) generate a cortex of correct size and with appropriate excitatory projection neuron diversity? While much progress has been made in the last decades, it is still unclear how intermediate progenitors (IPs) contribute to overall RGP output. To address this question, the authors established Rv-Flash Tag method that allows the labeling and tracing of individual units of clonally related progeny from RGPs. The advantage over existing technologies is the combination of clonal labeling with pulse labeling enabled by the Flash Tag technology. As such, the Flash Tag label is incorporated into dividing RGPs but it dilutes in a predictable and quantifiable manner in successive cell divisions. Thus, the lineage tree emerging from RGPs may be reconstructed with added information about intermediate amplification by IPs in a predictable manner. The authors collected a large data set that allows solid conclusions. The authors also used antibodies to define the major classes of excitatory projection neurons, and a battery of parameters to classify RGP clones (currently five classes of clones). Perhaps the most insightful analysis came from long-term live-imaging experiments whereby the authors could provide definitive and quantitative data regarding the rate of IP production by RGPs and how many times IPs divided to amplify clonal RGP output. Lastly, the authors established a model of RGP output which provides important information to complement existing experimental data.

Overall the study by Shen et al., provides interesting new RGP lineage data with added information about IP/IPP production. The authors extend previous models of RGP lineage progression and provide quantitative data that helps for a better understanding of the cellular mechanisms of RGP-mediated projection neuron generation during neocortical ontogeny. The data is of high quality, the manuscript well written and the illustrations are clear. Their data and proposed model at the RGP cell level will certainly inspire many future studies, especially at the molecular level, to decipher the involved mechanisms. Below are a few suggestions that may help to improve the reading experience:

1. In their discussion, the authors offer a few scenarios that may explain the heterogeneity of RGP lineages. While the listed scenarios are certainly valid explanations, the authors should add some more points of discussion. They should especially

consider the recent work by the laboratory of Josh Huang (Huilgol et al., 2023 Neuron; Huilgol et al., 2024 bioRxiv; Matho et al., 2021 Nature). In particular, the Matho publication is important since it proposes distinct classes of RGPs (albeit at population level). The Huilgol studies provide complementary information to the present study and should be mentioned in the discussion.

2. In previous work, by the Victor Tarabykin laboratory, they propose a feedback mechanism regulating progenitor and/or intermediate progenitor proliferation behavior. These studies should also be considered in the discussion.
3. Some of the findings in the present study are intriguing, especially in the light of previous work by the Borrell lab that found that direct neurogenesis in mammals are relatively minor. The authors should include a critical assessment and discussion regarding the discrepancies based on previous literature and new data.
4. Some expressions are a bit unusual and the authors may consider to revise, eg. 'stationary transition probability' which is used at a number of passages but its meaning is not completely clear.

Referee #2:

Referee comments

In this manuscript, the authors developed a new method for tracing the cell lineage in the developing mouse neocortex and examined the patterns of neurogenesis. The authors used a retrovirus and FlashTag (Rv-FlashTag) labeling technique for tracking the cells and their division. Using this method, the authors provided data complementing the previous knowledge and deepened our understanding of neocortical neurogenesis. The method is based on the assumption that the signal intensity of the FlashTag in a cell becomes half after one cell division. This assumption, however, might not be true in some cases. One example is shown in Figure 1G, H, K. n2 and n3 are the siblings, but the intensity is approximately 70% higher in n3 compared to n2. This result suggests that FlashTag can be distributed to the daughter cells unevenly. Such uneven distribution of FlashTag might be caused by the polarized morphology of radial glia cells. As the author showed in Figure EV1C, FlashTag signal is detected not only in the nuclei but also in the cytoplasm including the apical and basal processes. In the case of asymmetric division of the radial glia cells, one cell inherits a process and the other one does not, potentially resulting in uneven inheritance of FlashTag. Since the authors' assumption depicted in Figure 1J is the foundation of this manuscript, especially for the analysis in Figures 2 and 3, the authors need to address this concern. This can be addressed by the time-lapse imaging in combination with the FlashTag labeling.

In addition, the lineage reconstruction largely depends on the above-mentioned assumption and the localization of progenies. This reconstruction method may not reflect reality. For example, why is n3 considered as a progeny of G2, not G4? While the authors are well aware of this concern ("neuron selection in some lineage trees may not accurately reflect the actual neurogenic process"), they did not address this concern in the manuscript.

Minor comments

1. Page 8 line 1 The following sentence needs to be revised: "..were identified with either only 1 cell, or 2 or 4 cells with..".
2. Page 3 line 14 Cite DOI:10.1016/S0896-6273(01)00420-2 instead of Miyata et al., 2004.
3. The images of CSFE in Figure 1G and H are completely over exposed (e.g. n1 and n6), therefore, the signal intensity measurement should not be accurate.
4. How did the authors confirm the fate of the GFP+ cells in Figure 4?

Referee #3:

In the submitted manuscript, Shen et al. performed clonal lineage tracing of radial glial progenitors (RGPs) in developing mouse cerebral cortex. Although similar clonal analyses have been reported previously (e.g. Gao et al. Cell, 159, 775-788, 2014), the authors focused on a balance between direct production of neurons from RGPs (direct neurogenesis) and indirect neuron production via intermediate progenitors (indirect neurogenesis) by using a retrovirus and FlashTag (Rv-FlashTag) labeling technique. The authors showed a heterogeneity of RGPs and classified RGPs into five clusters mainly based on features of division patterns and future neuronal identities of the progenies.

From the results of a series of clonal lineage tracing experiments, the authors provide a new model of RGP division and its mediated neuronal production during cerebral cortical development; while RGPs equally generate all types of progenies (neurons and neurogenic and proliferative intermediate progenitors) in the first division, probabilities of transition of the subsequent RGP division patterns are relatively constant, and this model is supported by their mathematical model, which strengthens their conclusion.

Overall, although a large part of this manuscript is description, rather than elucidating molecular machinery, it would contribute to future molecular understanding of regulatory mechanisms of RGP division and brain size, whose disruption causes severe brain malformation such as microcephaly. However, this reviewer also found several issues remain to be addressed and several additional experiments are needed before I would recommend this manuscript for publication.

[Major points]

1) The authors mentioned that the final layer distribution of the neurons derived from the same IP is close in P7 cortex as previously reported, but the entire distribution patterns of the progenies of each RGP were not examined. Because five different RGP clusters are determined in this paper, the authors should examine spatial preference (rostral-caudal and medio-lateral localization) of each RGP cluster-derived neurons at P7.

2) In general, neurons born at E12 are located in the deep layers. However, almost all #4-RGPs only generate superficial layer neurons, whose final (neurogenic) division should occur at E15-E16. The IPP divisions (67.3% of #4-RGPs) may explain this inconsistency, because it may take a couple of days to divide several times. In case of the N divisions, however, #4-RGPs directly produce superficial layer neurons at E12 or E13, which may contradict previous reports (e.g. Takahashi et al. *J Neurosci*, 19, 10357-10371, 1999). The authors should explain this discrepancy. One possibility is that #4-RGPs transiently enter a dormant state (or very long G1 phase of the cell cycle) before direct neurogenesis. If the authors measure the cell cycle length of RGPs in the slice culture, you may be able to define whether a small part of RGPs shows a longer cell cycle length.

3) In line with my previous comment, it would be better to know the cell cycle length of proliferative and neurogenic intermediate progenitors, which may help to presume the timing of the final (neurogenic) division of neurons derived from the IP or IPP divisions.

4) The findings that IPP contributes to neuron production at the same level as direct neurogenesis are interesting. I wonder if the IPP includes outer Radial Glial progenitors (oRG), because previous studies indicate that oRGs repeatedly divide in the SVZ of mouse cerebral cortex (Wang et al. *Nat Neurosci*, 14, 555-561, 2011). The ratio of the IPP that possesses a pia-directed long process (a feature of oRGs) should be measured in the slice culture.

5) Previous reports indicate that a part of RGPs in late neurogenesis simultaneously produces two daughter neurons to reduce RGPs for finalization of neurogenesis (Miyata et al. *Development*, 131, 3133-3145, 2004; Takahashi et al. *J Neurosci*, 16, 6183-6196, 1996). On the other hand, late RGPs are also reported to transform into astrogenic progenitors after neurogenesis. In your clonal analyses, the ratio of the RGP division generating two daughter neurons should be measured.

6) The authors showed that pyramidal neurons derived from the N, IP and IPP divisions are positioned in an inside-out manner. Surely the inside-out cortical layering may partly be determined by the RGP properties. However, considering that neuronal migration speed after the multipolar-to-bipolar transition is relatively constant before a tip of the leading process reaches the marginal zone (Nadarajah et al. *Nat Neurosci*, 4, 143-150, 2001; Nishimura et al. *J Biol Chem*, 285, 5878-5887, 2010), it may also be important when neurons start migration. The length of time from the first division of RGPs to the migration start should be longer in the IPP divisions. The average duration before the migration start should be measured in the N, IP and IPP divisions by using the long-term time-lapse method.

[Minor points]

7) In Fig. EV4, it is hard to identify the glial cell number and morphology. Co-staining of the nuclei (e.g. DAPI) is required.

8) The authors mentioned that early-generation neurons are distributed broadly in the six-layered cerebral cortex, whereas late-generation neurons are restricted to locate in the superficial layers. Historically, this model has previously been proposed (Frantz et al. *Neuron*, 17, 55-61, 1996) and therefore the authors should cite this paper and discuss your revised model.

Point-by-point response to Reviewers' comments**Reviewer #1**

1. *“In their study, the authors addressed an important question in cortical development. How can radial glial progenitors (RGPs) generate a cortex of correct size and with appropriate excitatory projection neuron diversity? While much progress has been made in the last decades, it is still unclear how intermediate progenitors (IPs) contribute to overall RGP output. To address this question, the authors established Rv-Flash Tag method that allows the labeling and tracing of individual units of clonally related progeny from RGPs. The advantage over existing technologies is the combination of clonal labeling with pulse labeling enabled by the Flash Tag technology. As such, the Flash Tag label is incorporated into dividing RGPs but it dilutes in a predictable and quantifiable manner in successive cell divisions. Thus, the lineage tree emerging from RGPs may be reconstructed with added information about intermediate amplification by IPs in a predictable manner. The authors collected a large data set that allows solid conclusions. The authors also used antibodies to define the major classes of excitatory projection neurons, and a battery of parameters to classify RGP clones (currently five classes of clones). Perhaps the most insightful analysis came from long-term live-imaging experiments whereby the authors could provide definitive and quantitative data regarding the rate of IP production by RGPs and how many times IPs divided to amplify clonal RGP output. Lastly, the authors established a model of RGP output which provides important information to complement existing experimental data. Overall the study by Shen et al., provides interesting new RGP lineage data with added information about IP/IPP production. The authors extend previous models of RGP lineage progression and provide quantitative data that helps for a better understanding of the cellular mechanisms of RGP-mediated projection neuron generation during neocortical ontogeny. The data is of high quality, the manuscript well written and the illustrations are clear. Their data and proposed model at the RGP cell level will certainly inspire many future studies, especially at the molecular level, to decipher the involved mechanisms. Below are a few suggestions that may help to improve the reading experience.”*

Response: We sincerely thank the reviewer for the positive views and the constructive comments.

2. *“In their discussion, the authors offer a few scenarios that may explain the heterogeneity of RGP lineages. While the listed scenarios are certainly valid explanations, the authors should add some more points of discussion. They should especially consider the recent work by the laboratory of Josh Huang (Huilgol et al., 2023 Neuron; Huilgol et al., 2024 bioRxiv; Matho et al., 2021 Nature). In particular, the Matho publication is important since it proposes distinct classed of RGPs (albeit at population level). The Huilgol studies provide*

complementary information to the present study and should be mentioned in the discussion."

Response: We thank the reviewer for these valuable suggestions. In the revised manuscript, we have substantially expanded our discussion to incorporate the important recent work from Josh Huang's laboratory, which provides crucial complementary insights to our findings.

Regarding the Matho et al study (Matho *et al*, 2021): We have added discussion of this seminal work on page 18, lines 2-7: "*Using an intersection-subtraction strategy with Lhx2-CreER and Fezf2-Flp mouse lines, Matho et al. identified distinct RGP subpopulations in the dorsal pallium based on Lhx2 and Fezf2 expression patterns. Their lineage tracing revealed that RGP^{Lhx2+/Fezf2-} specifically generate callosal PyNs, while RGP^{Lhx2+/Fezf2+} produce subcortical PyNs, demonstrating fate-restricted RGP lineages generating distinct neuronal projection classes.*" This population-level evidence for RGP heterogeneity strongly supports our single-cell lineage tracing findings that demonstrate intrinsic diversity within individual RGP lineages.

Regarding the Huilgol et al. study (Huilgol *et al*, 2023): We have incorporated discussion of their quantitative findings on page 19, lines 5-12: "*The contribution of indirect neurogenesis also varies spatially across brain regions. Along the cortical medial-lateral axis, indirect neurogenesis makes progressively lower contributions, with sharp decreases in the amygdala and piriform cortex (Huilgol et al, 2023). Notably, direct and indirect neurogenesis differentially contribute to the generation of PyN projection types in the neocortex (Huilgol et al., 2023)*" This study demonstrates that indirect neurogenesis exhibits region-specific contributions, providing important spatial context for understanding the functional significance of the IP-mediated amplification we observe.

Regarding the Huilgol et al. study (Huilgol *et al*, 2025) : We have added detailed discussion on page 18, lines 28-29 and page19, lines 1-5: "*Using Tbr2-2A-CreER;Ai14 mice, Huilgol et al. revealed that IP-derived PyN laminar organization deviates substantially from conventional inside-out cortical development. Although the overall IP-derived PyN population follows a broad deep-to-superficial migration pattern, PyNs destined for 2-3 non-consecutive layers are produced simultaneously, while different PyN subtypes within individual layers are generated at distinct, non-contiguous developmental time points.*" This elegant work provides strong validation for our central conclusion that indirect neurogenesis serves dual functions: amplifying neuronal output while fundamentally expanding neuronal diversity through complex temporal-spatial production dynamics.

Together, these studies from the Huang laboratory provide essential population-level and temporal context that complements our single-cell lineage analysis, collectively demonstrating that cortical neurogenesis involves far greater complexity and heterogeneity than previously appreciated.

3. *“In previous work, by the Victor Tarabykin laboratory, they propose a feedback mechanism regulating progenitor and/or Intermediate progenitor proliferation behavior. These studies should also be considered in the discussion.”*

Response: We thank the reviewer for highlighting this important aspect of progenitor regulation. In the revised manuscript, we have incorporated discussion of the feedback mechanisms regulating RGP fate decisions proposed by the Victor Tarabykin laboratory and cited the relevant literatures (page 21, lines 8-10).

4. *“Some of the findings in the present study are intriguing, especially in the light of previous work by the Borrell lab that found that direct neurogenesis in mammals are relatively minor. The authors should include a critical assessment and discussion regarding the discrepancies based on previous literature and new data.”*

Response: We appreciate the reviewer's thoughtful observation regarding the apparent discrepancy between our findings and previous work from the Borrell lab and others suggesting minimal direct neurogenesis in mammals. In the revision, we have added a comprehensive paragraph discussing these discrepancies and their potential underlying causes (page 19, lines 26-29 and page 20, lines 1-15).

5. *“Some expressions are a bit unusual and the authors may consider to revise, eg. 'stationary transition probability' which is used at a number of passages but its meaning is not completely clear.”*

Response: In response to this comment, we have revised the manuscript to replace "stationary transition probability" with "stable transition probability" throughout the text.

Reviewer #2

1. *“In this manuscript, the authors developed a new method for tracing the cell lineage in the developing mouse neocortex and examined the patterns of neurogenesis. The authors used a retrovirus and FlashTag (Rv-FlashTag) labeling technique for tracking the cells and their division. Using this method, the authors provided data complementing the previous knowledge and deepened our understanding of neocortical neurogenesis.”*

Response: We sincerely thank the reviewer for the positive views and the constructive comments.

2. *“The method is based on the assumption that the signal intensity of the FlashTag in a cell becomes half after one cell division. This assumption, however, might not be true in some cases. One example is shown in Figure 1G, H, K. n2 and n3 are the siblings, but the intensity is approximately 70% higher in n3 compared to n2. This result suggests that FlashTag can be distributed to the daughter cells unevenly. Such uneven distribution of FlashTag might be caused by the polarized morphology of radial glia cells. As the author showed in Figure EV1C, FlashTag signal is detected not only in the nuclei but also in the cytoplasm including the apical and basal processes. In the case of asymmetric division of the radial glia cells, one cell inherits a process and the other one does not, potentially resulting in uneven inheritance of FlashTag. Since the authors' assumption depicted in Figure 1J is the foundation of this manuscript, especially for the analysis in Figures 2 and 3, the authors need to address this concern. This can be addressed by the time-lapse imaging in combination with the FlashTag labeling.”*

Response: We thank the reviewer for raising this important concern about the fundamental assumption underlying our FlashTag-based lineage tracing method. We acknowledge that uneven distribution of FlashTag between daughter cells could potentially affect our analysis, and we have conducted additional experiments and analysis to address this issue.

Uneven distribution of FlashTag in IP-generated daughter neurons

We indeed observed some degree of uneven FlashTag distribution between daughter cells produced by IPs. Using embryonic PH3 labeling and live cell fluorescence imaging, we found that the difference in FlashTag fluorescence between the soma of IP-generated two daughter cells was typically within 20% (**Fig. EV1E and EV1J**). This variability has a limited effect on determining daughter cell relationships and generational assignments within RGP lineages.

In the specific example shown in Figure 1, FlashTag fluorescence of n2 and n3 accounted for 37% and 63% of their combined total intensity, respectively—a difference of 26% (slightly higher than our embryonic observations but still within an acceptable range). Critically, when we examined the generational pattern, the

combined fluorescence of n2, n3, n4, and n5 in G2 totaled 190.3×10^7 fluorescence units, which closely matches the fluorescence of the single neuron n1 produced by IP in G1 (195.4×10^7) and is approximately twice the intensity of n6 in G3 (100.4×10^7). These quantitative relationships support the robustness of our generational analysis despite modest variability between individual daughter cells.

Influence of FlashTag in RGP processes

To address the concern about FlashTag distribution in radial glial processes potentially affecting our analysis, we conducted additional experiments to quantify the contribution of process-associated fluorescence. We injected low-titer RV-Cre and FlashTag into the lateral ventricle of E12 Ai140 mouse embryos and analyzed brain sections fixed at E14, followed by PFA fixation at E14 and subsequent brain sectioning, and then measured FlashTag fluorescence in individual lineages with only one first-generation daughter cell (**Fig. EV2A**). This approach enabled us to measure FlashTag distribution across different cellular compartments more precisely than time-lapse imaging in combination with the FlashTag labeling would allow. Through careful analysis of FlashTag fluorescence distribution in RGP soma, processes, and first-generation daughter cell (**Fig. EV2B-D**), we found that FlashTag in RGP processes accounted for only ~9.8% of the total RGP fluorescence (**Fig. EV2E**). Statistical analysis revealed that FlashTag fluorescence was roughly comparable between RGP soma and first-generation daughter cell (**Fig. EV2F**). Importantly, during asymmetric divisions, the fluorescence in RGP processes is not inherited by daughter cells. Instead, daughter cells inherit approximately half of the fluorescence from the mother RGP soma, while the remaining RGP retains the other half. Since the process-associated fluorescence (~9.8%) represents a small fraction that doesn't participate in inheritance patterns, it does not significantly affect our cell lineage reconstruction and analysis.

3. *“In addition, the lineage reconstruction largely depends on the above-mentioned assumption and the localization of progenies. This reconstruction method may not reflect reality. For example, why is n3 considered as a progeny of G2, not G4? While the authors are well aware of this concern (“neuron selection in some lineage trees may not accurately reflect the actual neurogenic process”), they did not address this concern in the manuscript.”*

Response: Our lineage reconstruction primarily relies on one of the principles is that IP-generated daughter neurons are spatial proximity in the cortex, which has been well-established in previous studies (Huilgol *et al.*, 2025; Mihalas & Hevner, 2018; Noctor *et al.*, 2004; Qian *et al.*, 1998). For instance, in Figure 1, we determined that the four neurons n2-n5 originated from IPP based on clustering patterns in cortical layer 5. Although n7 exhibits the same level of FlashTag fluorescence as n2-n5, n7 is spatially distant from the n2-n5 cluster and located in a more superficial cortical layer (presumably layer 4). Based on these spatial and laminar differences, we determined that n2-n5 originated from the IPP division in G2, whereas n7 belonged to the direct

neurogenic (N) division in G4. The close proximity of n2-n5 supports their shared IPP origin, while n7's isolated position and superficial location are consistent with a later neurogenic division.

In the original manuscript, we noted that "neuron selection in some lineage trees may not accurately reflect the actual neurogenic process" to illustrate the uncertainty associated with selecting neuron pairs originating from IPP divisions. As described above, when four IPP-originated neurons are closely clustered in the neocortex, the selecting two pairs of neurons based on spatial location relationships that originated from two different IPs produced by IPP division may not always accurately reflect the actual neurogenesis. To improve clarity, we have revised this statement to: "In particular, the selection of pair neurons originating from the IPs in the IPP division may not accurately reflect the neurogenic process. Given these methodological limitations, our subsequent statistical analyses emphasize generation-level comparisons rather than detailed pairwise lineage assignments."

In light of these methodological constraints, our downstream statistical analyses focus on generation-level analysis rather than detailed pair-wise assignments. This approach acknowledges the limitations of precise pair assignments while maintaining the validity of our generational analysis, which relies on the robust spatial clustering patterns and fluorescence intensity trends across generations. We emphasized these in revised manuscript (page 8, lines 4-8)

In conclusion, our lineage reconstruction strategy combines established spatial proximity principles with fluorescence intensity patterns to provide the most accurate possible reconstruction given the technical constraints. While we acknowledge that individual neuron pair assignments may contain uncertainties, the overall generational patterns and spatial clustering relationships provide robust support for our conclusions about cortical neurogenesis and lineage progression.

4. *"Page 8 line 1 The following sentence needs to be revised: "..were identified with either only 1 cell, or 2 or 4 cells with.."."*

Response: We have revised it accordingly (page 8, lines 12-13).

5. *"Page 3 line 14 Cite DOI:10.1016/S0896-6273(01)00420-2 instead of Miyata et al., 2004."*

Response: We have updated the citation to DOI:10.1016/S0896-6273(01)00420-2 as suggested.

6. *"The images of CSFE in Figure 1G and H are completely over exposed (e.g. n1 and n6), therefore, the signal intensity measurement should not be accurate."*

Response: We thank the reviewer for the attention to the technical aspects of our imaging methodology. To minimize variations and prevent overexposure, we

employed highly standardized methodologies throughout our pipeline, from RV-Cre and CFSE injection to imaging and reconstruction.

For imaging, we implemented a specific protocol to avoid overexposure: we first captured neurons with the highest fluorescence intensity in each lineage, carefully adjusting parameters to ensure they were not overexposed. We then used these same parameters to photograph all other neurons in the same lineage, maintaining accurate relative intensity measurements.

We have double-checked our original images and confirmed that n1 and n6 were not overexposed. The fluorescence signals retain sufficient detail for accurate intensity quantification within our imaging system's dynamic range.

We emphasize this imaging process in the Methods section (page 23, line 17-29 and page 24, lines 1-9) and have provided all source data files with the revised manuscript.

7. *“How did the authors confirm the fate of the GFP+ cells in Figure 4?”*

Response: We identified three distinct fate outcomes based on the following criteria:

Direct Neurogenesis: We identified direct neurogenesis when a postmitotic cell was generated by RGP division, no longer divided, and migrated away from the SVZ. This represents the direct production of neurons without intermediate progenitor stages.

IP Division: We confirmed IP division when RGP divisions resulted in asymmetric fate outcomes producing intermediate progenitors (IPs) that subsequently generated two daughter cells and migrated away from the SVZ.

IPP Division: IPP divisions were identified when an RGP division gave rise to a proliferative intermediate progenitor (IPP), which was confirmed by observing its symmetric division in the SVZ to generate two IPs. These IPs then underwent further symmetric divisions to produce four postmitotic neurons, which subsequently migrated away from the SVZ. In some cases, only one of the two IPs was observed to generate two neurons at the time of imaging.

Reviewer #3

1. *“In the submitted manuscript, Shen et al. performed clonal lineage tracing of radial glial progenitors (RGPs) in developing mouse cerebral cortex. Although similar clonal analyses have been reported previously (e.g. Gao et al. Cell, 159, 775-788, 2014), the authors focused on a balance between direct production of neurons from RGPs (direct neurogenesis) and indirect neuron production via intermediate progenitors (indirect neurogenesis) by using a retrovirus and FlashTag (Rv-FlashTag) labeling technique. The authors showed a heterogeneity of RGPs and classified RGPs into five clusters mainly based on features of division patterns and future neuronal identities of the progenies. From the results of a series of clonal lineage tracing experiments, the authors provide a new model of RGP division and its mediated neuronal production during cerebral cortical development; while RGPs equally generate all types of progenies (neurons and neurogenic and proliferative intermediate progenitors) in the first division, probabilities of transition of the subsequent RGP division patterns are relatively constant, and this model is supported by their mathematical model, which strengthens their conclusion. Overall, although a large part of this manuscript is description, rather than elucidating molecular machinery, it would contribute to future molecular understanding of regulatory mechanisms of RGP division and brain size, whose disruption causes severe brain malformation such as microcephaly. However, this reviewer also found several issues remain to be addressed and several additional experiments are needed before I would recommend this manuscript for publication.”*

Response: We sincerely thank the reviewer for the positive views and the constructive comments.

2. *“The authors mentioned that the final layer distribution of the neurons derived from the same IP is close in P7 cortex as previously reported, but the entire distribution patterns of the progenies of each RGP were not examined. Because five different RGP clusters are determined in this paper, the authors should examine spatial preference (rostral-caudal and medio-lateral localization) of each RGP cluster-derived neurons at P7.”*

Response: We thank the reviewer for this important suggestion to examine the spatial preferences of five RGP lineage clusters. We have conducted a comprehensive spatial distribution analysis as requested.

Due to the difficulty of precisely distinguishing rostral-caudal and medial-lateral areas of the neocortex, we adopted a functional area-based approach for our spatial analysis. We compared the five RGP lineage clusters across four distinct cortical regions: motor cortex (MC), somatosensory cortex (SC), auditory cortex (AC), and visual cortex (VC). These four cortices tend to be located rostral, medial, lateral, and caudal to the brain, respectively, providing effective spatial coverage for our analysis. The location

of each lineage in the neocortex was accurately verified using the Allen Mouse Brain Atlas, ensuring precise anatomical classification of our samples.

Our analysis revealed that the five RGP lineage clusters exhibited distinct spatial distributions across the neocortex (**Fig. EV6A**). This spatial heterogeneity suggests that different RGP lineage clusters have region-specific preferences in their distribution patterns. For example, cluster 2 was significantly enriched in the auditory cortex (AC) compared to the motor cortex (MC), somatosensory cortex (SC), and visual cortex (VC). In contrast, cluster 4 showed more prominent representation in the motor cortex (MC) than in the other three regions. Notably, neither clone size nor generation number showed significant differences across the four cortical areas (**Fig. EV6B,C**).

3. *“In general, neurons born at E12 are located in the deep layers. However, almost all #4-RGPs only generate superficial layer neurons, whose final (neurogenic) division should occur at E15-E16. The IPP divisions (67.3% of #4-RGPs) may explain this inconsistency, because it may take a couple of days to divide several times. In case of the N divisions, however, #4-RGPs directly produce superficial layer neurons at E12 or E13, which may contradict previous reports (e.g. Takahashi et al. J Neurosci, 19, 10357-10371, 1999). The authors should explain this discrepancy. One possibility is that #4-RGPs transiently enter a dormant state (or very long G1 phase of the cell cycle) before direct neurogenesis. If the authors measure the cell cycle length of RGPs in the slice culture, you may be able to define whether a small part of RGPs shows a longer cell cycle length.”*

Response: We thank the reviewer for raising this important question about the apparent discrepancy between birth timing and layer distribution in #4-RGPs. We systematically analyzed all lineages in the #4-cluster and identified a total of 9 lineages where the first generation (G1) of RGP undergoes direct neurogenesis (N division) (**Appendix Figure**).

Among the 9 lineages with direct neurogenesis in G1, 7 lineages had G1 neurons localized in layer 6 or the layer 5/6 junction region of the neocortex (marked by red solid arrows). The remaining 2 neurons in G1 were located in neocortical layer 5 (marked by red open arrows). These results are consistent with previous studies showing that early-born neurons are mainly distributed in the deep layers of the neocortex.

To further investigate the reviewer's suggestion about potential dormant states or extended cell-cycle length, we analyzed our time-lapse imaging data for cell cycle characteristics. We found that the average cell cycle length of N division of RGPs was 18.08 ± 1.74 hours, which fell within the expected range for RGP divisions during this developmental period (Miyata *et al*, 2004; Shitamukai *et al*, 2011). However, there was significant heterogeneity in cell cycle duration, with a maximum length of 36 hours. This result suggests that we cannot exclude the possibility that some RGPs

transiently enter a dormant state (or very long cell cycle length) before direct neurogenesis.

Appendix Figure

4. “In line with my previous comment, it would be better to know the cell cycle length of proliferative and neurogenic intermediate progenitors, which may help to presume the timing of the final (neurogenic) division of neurons derived from the IP or IPP divisions.”

Response: We thank the reviewer for this excellent suggestion to examine cell cycle lengths of proliferative and neurogenic intermediate progenitors. This analysis provides important insights into the timing of neurogenic divisions and helps explain the final positioning of neurons derived from different division patterns.

We examined how cell cycle duration influenced PyN birth time across different division patterns. Our time-lapse imaging analysis revealed the average cell cycle lengths were 18.08 ± 1.74 hours for RGPs generating postmitotic neurons directly (t_1), 15.00 ± 4.60 hours for RGPs producing IPs (t_2), and 16.95 ± 2.75 hours for IPs generating postmitotic neurons (t_3). There were no significant differences among these three groups (**Fig. 4I,J**), indicating that individual division steps maintain relatively consistent timing regardless of the division pattern. In contrast, the total time required for IPPs to produce postmitotic neurons after undergoing two rounds of divisions was significantly extended to 32.75 ± 5.24 hours (**Fig. 4I,J**). This duration is nearly double that of direct IP neurogenesis (t_3) and reflects the cumulative time required for the additional division round (**Fig. 4I,J**).

These results suggest that additional rounds of division via IPs/IPP delay the birthdate of PyNs upon RGP division, potentially affecting their final radial positioning in the neocortex.

5. *“The findings that IPP contributes to neuron production at the same level as direct neurogenesis are interesting. I wonder if the IPP includes outer Radial Glial progenitors (oRG), because previous studies indicate that oRGs repeatedly divide in the SVZ of mouse cerebral cortex (Wang et al. Nat Neurosci, 14, 555-561, 2011). The ratio of the IPP that possesses a pia-directed long process (a feature of oRGs) should be measured in the slice culture.”*

Response: We thank the reviewer for this important question about the potential contribution of outer radial glial progenitors (oRGs) to our IPP population.

The study by Wang et al. revealed that oRGs represented only 5–7% of all radial glial progenitors (RGPs) and were relatively rare at all stages of mouse embryonic development (Wang *et al.*, 2011). This low prevalence suggests that oRGs would constitute a minor fraction of the progenitor population during our experimental timeframe.

Similar to IPs, oRGs are also originated from RGPs, but they have distinct morphological characteristics that allow for their identification. oRGs are characterized by a distinctive morphology featuring a long basal process extending toward the pial surface but notably lacking an apical process. In contrast, IPs typically exhibit a multipolar morphology with short processes. They do not possess the long basal fiber characteristic of oRGs.

Based on our comprehensive live imaging analysis, we did not observe the generation of oRG-like cells from RGPs in any of the 74 recorded divisions. This has been noted in the revised manuscript (page 13, lines 9-10).

6. *“Previous reports indicate that a part of RGPs in late neurogenesis simultaneously produces two daughter neurons to reduce RGPs for finalization of neurogenesis (Miyata et al. Development, 131, 3133-3145, 2004; Takahashi et al. J Neurosci, 16, 6183-6196, 1996). On the other hand, late RGPs are also reported to transform into astrogenic progenitors after neurogenesis. In your clonal analyses, the ratio of the RGP division generating two daughter neurons should be measured.”*

Response: Our analysis revealed that that ~14% of lineages involved RGP terminal divisions, in which two daughter neurons were generated simultaneously (**Fig. 2G**).

7. *“The authors showed that pyramidal neurons derived from the N, IP and IPP divisions are positioned in an inside-out manner. Surely the inside-out cortical layering may partly be determined by the RGP properties. However, considering that neuronal migration speed after the multipolar-to-bipolar transition is relatively constant before a tip of the leading process reaches the marginal zone (Nadarajah et al. Nat Neurosci, 4, 143-150, 2001; Nishimura et al. J Biol Chem, 285, 5878-5887, 2010), it may also be important when neurons start migration. The length of time from the first division of RGPs to the migration start should be*

longer in the IPP divisions. The average duration before the migration start should be measured in the N, IP and IPP divisions by using the long-term time-lapse method.”

Response: We thank the reviewer for this insightful comment about the potential role of migration timing in cortical layering. We measured the residence time of postmitotic neurons produced by different division patterns in the SVZ using our long-term time-lapse imaging approach. Specifically, we quantified the time between a neuron's production and its departure from the SVZ, which represented the duration before migration initiation for neurons derived from RGPs (N division), IPs (IP division), and IPPs (IPP division). We found that the average residence time were 28.90 ± 2.80 hours for N-division neurons (T_1), 4.33 ± 4.32 hours for IP-division neurons (T_2), and 29.90 ± 4.27 hours for IPP-division neurons (T_3) (**Fig. 4I,J**). Although substantial numerical variation was observed within each group, there were no statistically significant differences between the three groups (**Fig. 4I,J**). These results suggest that the inside-out layering positioning is primarily determined by the birth timing differences to each division pattern rather than differential residence time.

8. *“In Fig. EV4, it is hard to identify the glial cell number and morphology. Co-staining of the nuclei (e.g. DAPI) is required.”*

Response: To minimize experimental variations and ensure reproducibility, we employed highly standardized methodologies throughout our entire analytical pipeline, from CFSE injection and fluorescent immunostaining to imaging and reconstruction procedures.

Regarding the immunostaining protocol, all brain slices were simultaneously labeled with a comprehensive panel of 4 antibodies and 1 dye, including GFP, Satb2, Ctip2, Fog2, and CFSE. Due to the constraints of our multi-channel imaging system, we did not have additional channels available for DAPI nuclear staining.

To address the reviewer's valid concern about glial cell visualization, we have enhanced **Fig. EV5** in the revised manuscript by providing high-resolution 3D images of glial cells. These 3D representations clearly demonstrate both the number and detailed morphology of glial cells.

9. *“The authors mentioned that early-generation neurons are distributed broadly in the six-layered cerebral cortex, whereas late-generation neurons are restricted to locate in the superficial layers. Historically, this model has previously been proposed (Frantz et al. Neuron, 17, 55-61, 1996) and therefore the authors should cite this paper and discuss your revised model.”*

Response: We have cited the paper and discuss our model in revised manuscript (page 18, and lines 17-25).

References

- Huilgol D, Levine JM, Galbavy W, Wang BS, He M, Suryanarayana SM, Huang ZJ (2023) Direct and indirect neurogenesis generate a mosaic of distinct glutamatergic projection neuron types in cerebral cortex. *Neuron* 111: 2557-2569.e2554
- Huilgol D, Levine JM, Galbavy W, Wang BS, Huang ZJ (2025) Orderly specification and precise laminar deployment of mouse cortical projection neuron types through intermediate progenitors. *Dev Cell* doi: 10.1016/j.devcel.2025.02.009
- Matho KS, Huilgol D, Galbavy W, He M, Kim G, An X, Lu J, Wu P, Di Bella DJ, Shetty AS *et al* (2021) Genetic dissection of the glutamatergic neuron system in cerebral cortex. *Nature* 598: 182-187
- Mihalas AB, Hevner RF (2018) Clonal analysis reveals laminar fate multipotency and daughter cell apoptosis of mouse cortical intermediate progenitors. *Development* 145
- Miyata T, Kawaguchi A, Saito K, Kawano M, Muto T, Ogawa M (2004) Asymmetric production of surface-dividing and non-surface-dividing cortical progenitor cells. *Development* 131: 3133-3145
- Noctor SC, Martínez-Cerdeño V, Ivic L, Kriegstein AR (2004) Cortical neurons arise in symmetric and asymmetric division zones and migrate through specific phases. *Nat Neurosci* 7: 136-144
- Qian X, Goderie SK, Shen Q, Stern JH, Temple S (1998) Intrinsic programs of patterned cell lineages in isolated vertebrate CNS ventricular zone cells. *Development* 125: 3143-3152
- Shitamukai A, Konno D, Matsuzaki F (2011) Oblique radial glial divisions in the developing mouse neocortex induce self-renewing progenitors outside the germinal zone that resemble primate outer subventricular zone progenitors. *J Neurosci* 31: 3683-3695
- Wang X, Tsai JW, LaMonica B, Kriegstein AR (2011) A new subtype of progenitor cell in the mouse embryonic neocortex. *Nat Neurosci* 14: 555-561

Dear Prof. Yu,

Thank you for submitting your revised manuscript (EMBOJ-2025-120450R) to The EMBO Journal for our consideration, and for your patience during peer review. Your manuscript has been sent back to the three original referees that had previously reviewed the initial version of your manuscript, and we have now received their comments, which you can find below.

I am pleased to say that the referees acknowledge that the majority of the initially raised concerns have been adequately addressed, and that the manuscript has been significantly strengthened by the addition of new data and analyses. Referees #1 and #3 have no further comments and now support publication of the manuscript. Referee #2, however, identifies a number of remaining concerns regarding lineage reconstruction, which in their view have not yet been sufficiently addressed or explained.

In light of this input, I would like to invite you to submit a final version of your manuscript fully addressing the five points made by referee #2, along with a detailed point-by-point response to their comments detailing any changes to the manuscript.

From the editorial side, there are also a few other changes and corrections we need you to make in the final version of the manuscript:

- Please note that all co-corresponding authors must provide their ORCID IDs; Lin-Yun Liu and Ying Zhu have not linked their ORCID IDs to their profiles yet.
- The author contributions statement should be removed from the manuscript file. Instead, we use CRediT to specify the contributions of each author in the journal submission system. Please feel free to use the free text box to provide more detailed descriptions during submission. See also our guide to authors for more information:
<https://www.embopress.org/page/journal/14602075/authorguide#authorshipguidelines>.
- Main and Expanded View (EV) Figures should be uploaded as individual, high-resolution Figures; the Figure legends should be placed below the References list of the main manuscript file.
- Please note that EMBO press papers are accompanied online by:
 - A) a short (2 sentences) summary of the findings and their significance,
 - B) 2-5 short bullet points highlighting the key results, and
 - C) a synopsis image in .jpg or .png format that is exactly 550 pixels wide and 300-600 pixels high (the height is variable). Please note that all text in the image needs to be legible at the final size.Please upload this information along with your revised manuscript (the text for A and B should be provided in a separate Word file).
- During our routine data checks, our data editors have raised the following queries regarding figures, data, and legends. Please make sure that all requests below are completely addressed in the final version of your manuscript (please highlight all changes in the manuscript):
 1. Please provide the exact p-values in the legends of Figures 3H, 5E.
 2. Please note that information related to "n" is missing in the legends of Figures 2B, C ; 3G, H; 4J, L; 5G-J; EV1 H, EV2 E, F; EV5 B, C.
- Table EV1 should be uploaded as an individual file.
- The legends of the Movies should be removed from the main manuscript file (and EV Figure file) and instead zipped together with each Movie file.

Please also note that as part of the EMBO publications' Transparent Editorial Process, The EMBO Journal publishes online a Peer Review File along with each accepted manuscript. This File will be published in conjunction with your paper and will include the referee reports, your point-by-point response and all pertinent correspondence relating to the manuscript. You can opt out of this by letting the editorial office know (contact@embojournal.org). If you do opt out, the Peer Review File link will point to the following statement: "No Peer Review File is available with this article, as the authors have chosen not to make the review process public in this case."

We look forward to seeing a final version of your manuscript as soon as possible. Please let us know if you have any questions and use this link to submit your revision: <https://emboj.msubmit.net/cgi-bin/main.plex>.

Best regards,

Ioannis

Referee #1:

The authors have addressed all my points from the initial review very well, and the manuscript has improved accordingly. I still think that the new findings and conclusions provide quite important information for the field. The manuscript should thus be reported soon.

Referee #2:

The authors tried to explain the solidness of their analysis about the lineage reconstruction, but there are still some critical concerns.

1. Even though the authors stated as "slightly higher than our embryonic observations but still within an acceptable range", the reviewer does not find any logical (statistical) explanation that the observed differences in the CSFE signal are "within acceptable range".
2. How did the authors take such uneven distribution of CSFE in daughter cells into consideration for the analysis?
3. Although the authors stated that they used spatial information of progenies, such information has not been clearly presented in the figures or tables. For example, what are the distances between cells shown in the Fig 1F?
4. The authors need to show that the CSFE signal used in the analysis is not saturated.
5. Fig 4: The authors need to show that the fates of the progenies by immunostaining after the time-lapse imaging, as previously done elsewhere (e.g. <https://doi.org/10.1038/nature08845>).

Referee #3:

In the revised version of the manuscript, the authors have adequately addressed my previous concerns by providing many additional data, including a spatial distribution pattern of each RGP lineage cluster, as well as detailed analyses of the cell cycle length and the departure timing of intermediate progenitors. I commend the authors for their substantial efforts in conducting new experiments. The manuscript has significantly been improved and now suitable for publication in EMBO Journal.

Point-by-point response to Editor and Reviewer comments**Editor**

1. *“Please note that all co-corresponding authors must provide their ORCID IDs; Lin-Yun Liu and Ying Zhu have not linked their ORCID IDs to their profiles yet.”*

Response: We have now linked the ORCID IDs for both Lin-Yun Liu and Ying Zhu to their respective profiles as required. All co-corresponding authors now have their ORCID IDs properly associated with their accounts.

2. *“The author contributions statement should be removed from the manuscript file. Instead, we use CRediT to specify the contributions of each author in the journal submission system. Please feel free to use the free text box to provide more detailed descriptions during submission. See also our guide to authors for more information:
<https://www.embopress.org/page/journal/14602075/authorguide#authorshipguidelines>.”*

Response: Thank you for this guidance. We have removed the author contributions statement from the manuscript file as requested. We have used the CRediT system to specify each author's contributions through the journal submission system, utilizing the free text box to provide detailed descriptions of individual contributions as needed. We have also reviewed the author guide at the provided link to ensure compliance with the journal's authorship guidelines.

3. *“Main and Expanded View (EV) Figures should be uploaded as individual, high-resolution Figures; the Figure legends should be placed below the References list of the main manuscript file.”*

Response: Thank you for this instruction. We have uploaded all Main and Expanded View Figures as individual, high-resolution files in PDF format. Additionally, we have moved all Figure legends to below the References list in the main manuscript file in accordance with the journal's formatting requirements.

4. *“A) a short (2 sentences) summary of the findings and their significance, B) 2-5 short bullet points highlighting the key results, and C) a synopsis image in .jpg or .png format that is exactly 550 pixels wide and 300-600 pixels high (the height is variable). Please note that all text in the image needs to be legible at the final size. Please upload this information along with your revised manuscript (the text for A and B should be provided in a separate Word file)”*

Response: We have prepared and uploaded all the requested materials.

5. *“Please provide the exact p-values in the legends of Figures 3H, 5E.”*

Response: We have updated the figure legends for Figures 3H and 5E to include the

exact *p*-values as requested.

6. "Please note that information related to "n" is missing in the legends of Figures 2B, C ; 3G, H; 4J, L; 5G-J; EV1 H, EV2 E, F; EV5 B, C."

Response: Thank you for pointing this out. We have added the exact sample size information (n numbers) to the legends of all the specified figures.

7. "*Table EV1 should be uploaded as an individual file.*"

Response: We have uploaded EV Tables as individual files.

8. "*The legends of the Movies should be removed from the main manuscript file (and EV Figure file) and instead zipped together with each Movie file.*"

Response: Thank you for this instruction. We have removed the movie legends from the main manuscript file and EV Figure file as requested. The movie legends have been zipped together with their corresponding movie files and uploaded separately.

9. "*Please also note that as part of the EMBO publications' Transparent Editorial Process, The EMBO Journal publishes online a Peer Review File along with each accepted manuscript. This File will be published in conjunction with your paper and will include the referee reports, your point-by-point response and all pertinent correspondence relating to the manuscript. You can opt out of this by letting the editorial office know (contact@embojournal.org). If you do opt out, the Peer Review File link will point to the following statement: "No Peer Review File is available with this article, as the authors have chosen not to make the review process public in this case."*"

Response: Thank you for informing us about The EMBO Journal's Transparent Editorial Process. We accept the publication of the Peer Review File online.

Reviewer #1

“The authors have addressed all my points from the initial review very well, and the manuscript has improved accordingly. I still think that the new findings and conclusions provide quite important information for the field. The manuscript should thus be reported soon.”

Response: We sincerely thank Reviewer #1 for the positive feedback and for acknowledging that we have addressed all the points from the initial review satisfactorily. We greatly appreciate his/her recommendation for publication and his/her constructive input throughout the review process, which has helped strengthen our work.

Reviewer #2

1. *“The authors tried to explain the solidness of their analysis about the lineage reconstruction, but there are still some critical concerns. Even though the authors stated as “slightly higher than our embryonic observations but still within an acceptable range”, the reviewer does not find any logical (statistical) explanation that the observed differences in the CFSE signal are “within acceptable range”. How did the authors take such uneven distribution of CFSE in daughter cells into consideration for the analysis?”*

Response: We thank the reviewer for this important question regarding the statistical basis for our analysis of CFSE fluorescence distribution deviations.

Indeed, CFSE fluorescence does not distribute exactly equally between daughter cells during each division, but instead exhibited relative deviations. In the revised manuscript, we systematically analyzed the deviation values of CFSE fluorescence between the two daughter cells in each division across all experimental lineages. We found that the normalized deviation values followed a normal distribution with mean (μ) = 0, standard deviation (σ) = 0.0541, and 99% confidence interval = ± 0.139 (**Fig. EV5A**). To rigorously assess whether these experimentally observed deviations compromise lineage reconstruction accuracy, we performed computational validation using 100 simulated lineage trees. Each simulation incorporated randomly sampled deviations from our experimentally characterized normal distribution (**Fig. EV5B-D**). Quantitative analysis revealed that all perturbed lineage trees were reconstructed with perfect accuracy, maintaining identical structure to their unperturbed counterparts (**Fig. EV5E,F**). These results demonstrate that the observed biological deviations in CFSE fluorescence distribution do not compromise the accuracy of automated lineage tree reconstruction.

Our revised analysis now provides the logical and statistical explanation that the reviewer correctly identified as missing. The observed CFSE distribution deviations are “within acceptable range” specifically because: (1) they follow a well-characterized normal distribution with quantified parameters derived from experimental data, and (2) computational validation demonstrates that deviations within this experimentally observed range do not affect reconstruction accuracy. This evidence-based approach replaces our previous subjective assessment with rigorous quantitative validation.

2. *“Although the authors stated that they used spatial information of progenies, such information has not been clearly presented in the figures or tables. For example, what are the distances between cells shown in the Fig 1F?”*

Response: We thank the reviewer for highlighting this important aspect of neuronal spatial information. In our revised manuscript, we have provided the distances between neurons shown in Figure 1F in **Table EV1**. This table includes pairwise distances measured in micrometers for all labeled neurons, allowing readers to assess

the spatial relationships within and between lineages.

3. “The authors need to show that the CFSE signal used in the analysis is not saturated.”

Response: We thank the reviewer for this important concern regarding potential CFSE signal saturation. We have conducted a comprehensive quantitative analysis to demonstrate that our CFSE signals are not saturated and are suitable for accurate quantitative analysis. All imaging was performed using a spinning-disk confocal microscope with a 60× oil-immersion objective and 16-bit acquisition mode (z-step size 0.33 μm), providing a dynamic range of 0–65,535 grayscale levels ($2^{16}-1$).

Using Imaris software, we analyzed all CFSE fluorescence intensity values from n1 and n7 in Figure 1G-H. For n6, CFSE fluorescence ranged from 5,506 to 61,278, demonstrating no saturation (**Appendix Figure A**). For n1, a very small subset of pixels exceeded 65,535 (**Appendix Figure B**, red arrows), but the saturated region occupied only 0.0522 μm³, representing merely 0.02% of the total somatic volume. The minimal saturation in n1 (0.02% of somatic volume) does not compromise quantitative analysis, as >99.98% of the CFSE signal remains within the dynamic range. These results demonstrate that our CFSE imaging conditions are appropriate for accurate lineage reconstruction and quantitative fluorescence analysis.

Appendix Figure

4. “Fig 4: The authors need to show that the fates of the progenies by immunostaining after the time-lapse imaging, as previously done elsewhere (e.g. <https://doi.org/10.1038/nature08845>).”

Response: We appreciate the reviewer's suggestion regarding post hoc immunohistochemical validation of cell fates following time-lapse imaging. We fully acknowledge that post hoc immunohistochemical validation represents the most direct approach to determine cell fate identity and recognize the value of this experimental approach as demonstrated in the referenced study.

Unfortunately, we did not preserve the cultured brain slices following the long-term time-lapse imaging experiments, making this specific validation experiment technically unfeasible under our current experimental conditions. We recognize this as a limitation of our current study design.

However, we would like to emphasize that our approach of using cell differentiation and migration trajectories to determine daughter cell fates is well-established and widely accepted in the field, as demonstrated in published studies (Nishimura *et al*, 2010; Noctor *et al*, 2004; Noctor *et al*, 2008; Zhao *et al*, 2020). This methodology has been successfully employed to track lineage progression and identify cell fate decisions based on morphological characteristics, division patterns, and migratory behavior during development.

We will consider incorporating post hoc immunohistochemical validation in future experiments to strengthen our experimental approach and provide even more robust evidence for lineage tracing conclusions.

Reference

Nishimura YV, Sekine K, Chihama K, Nakajima K, Hoshino M, Nabeshima Y, Kawauchi T (2010) Dissecting the factors involved in the locomotion mode of neuronal migration in the developing cerebral cortex. *J Biol Chem* 285: 5878-5887

Noctor SC, Martínez-Cerdeño V, Ivic L, Kriegstein AR (2004) Cortical neurons arise in symmetric and asymmetric division zones and migrate through specific phases. *Nat Neurosci* 7: 136-144

Noctor SC, Martínez-Cerdeño V, Kriegstein AR (2008) Distinct behaviors of neural stem and progenitor cells underlie cortical neurogenesis. *J Comp Neurol* 508: 28-44

Zhao YF, He XX, Song ZF, Guo Y, Zhang YN, Yu HL, He ZX, Xiong WC, Guo W, Zhu XJ (2020) Human antigen R-regulated mRNA metabolism promotes the cell motility of migrating mouse neurons. *Development* 147

Reviewer #3

“In the revised version of the manuscript, the authors have adequately addressed my previous concerns by providing many additional data, including a spatial distribution pattern of each RGP lineage cluster, as well as detailed analyses of the cell cycle length and the departure timing of intermediate progenitors. I commend the authors for their substantial efforts in conducting new experiments. The manuscript has significantly been improved and now suitable for publication in EMBO Journal.”

Response: We thank the reviewer for the positive evaluation of our revised manuscript. We appreciate his/her recognition of the additional experiments and analyses we conducted to address his/her previous concerns. We are pleased that reviewer now find the manuscript suitable for publication in EMBO Journal.

Dear Prof. Yu,

Congratulations on an excellent manuscript! I am very pleased to inform you that it has been accepted for publication in The EMBO Journal. Thank you for comprehensively addressing the initially raised referee criticisms and the editorial requests for corrections and changes.

If you have any questions, please do not hesitate to contact the Editorial Office. Thank you for your contribution to The EMBO Journal. Working with you has been a pleasure.

Best regards,

Ioannis
